# Identification of drug-specific public TCR driving severe cutaneous adverse reactions

Ren-You Pan[1], Mu-Tzu Chu[2,3], Chuang-Wei Wang[1,2], Yun-Shien Lee[4], Francois Lemonnier [5], Aaron W. Michels[6], Ryan Schutte[7], David A. Ostrov [7], Chun-Bing Chen [1,2,8], Elizabeth Jane Phillips [9,10], Simon Alexander Mallal[9,10], Maja Mockenhaupt[11], Teresa Bellón [12], Wichittra Tassaneeyakul [13], Katie D. White[9], Jean-Claude Roujeau[14], Wen-Hung Chung [1,2,8,15,16] & Shuen-Iu Hung [2,3]

Drug hypersensitivity such as severe cutaneous adverse reactions (SCAR), including Stevens–Johnson syndrome (SJS) and toxic epidermal necrolysis (TEN), could be life-threatening. Here, we enroll SCAR patients to investigate the T cell receptor (TCR) repertoire by next-generation sequencing. A public αβTCR is identified from the cytotoxic T lymphocytes of patients with carbamazepine-SJS/TEN, with its expression showing drug/phenotype-specificity and an bias for HLA-B*15:02. This public αβTCR has binding affinity for carbamazepine and its structural analogs, thereby mediating the immune response. Adoptive transfer of T cell expressing this public αβTCR to *HLA-B*15:02* transgenic mice receiving oral administration of carbamazepine induces multi-organ injuries and symptoms mimicking SCAR, including hair loss, erythema, increase of inflammatory lymphocytes in the skin and blood, and liver and kidney dysfunction. Our results not only demonstrate an essential role of TCR in the immune synapse mediating SCAR, but also implicate potential clinical applications and development of therapeutics.

[1] Department of Dermatology, Drug Hypersensitivity Clinical and Research Center, Chang Gung Memorial Hospital, Linkou, Taipei, Keelung, Taoyuan 333, Taiwan. [2] Cancer Vaccine and Immune Cell Therapy Core Laboratory, Chang Gung Immunology Consortium, Chang Gung Memorial Hospital, Linkou, Taoyuan 333, Taiwan. [3] Institute of Pharmacology, National Yang-Ming University, Taipei 112, Taiwan. [4] Department of Biotechnology, Ming Chuan University, Taoyuan 333, Taiwan. [5] INSERM U1016, Institut Cochin, Equipe Immunologie du Diabète, Hôpital Saint-Vincent-de-Paul, 75674 Paris, Cedex 14, France. [6] Barbara Davis Center for Childhood Diabetes, University of Colorado Denver, Aurora, CO 80204, USA. [7] Department of Pathology, Immunology and Laboratory Medicine, University of Florida College of Medicine, Gainesville, FL 32611, USA. [8] College of Medicine, Chang Gung University, Taoyuan 333, Taiwan. [9] Departments of Medicine and Pathology, Microbiology and Immunology, School of Medicine, Vanderbilt University, Nashville, TN 37235, USA. [10] Institute for Immunology and Infectious Diseases, Murdoch University, Perth 6150 WA, Australia. [11] Dokumentationszentrum schwerer Hautreaktionen (dZh), Department of Dermatology, Medical Center and Medical Faculty, University of Freiburg, Freiburg 79085, Germany. [12] Research Unit, Hospital Universitario La Paz-Idi PAZ, Madrid 28046, Spain. [13] Department of Pharmacology, Faculty of Medicine, Khon Kaen University, Khon Kaen 40002, Thailand. [14] Emeritus Professor of Dermatology, Université Paris-Est Créteil (UPEC), Créteil 94000, France. [15] Department of Dermatology, Xiamen Chang Gung Hospital, Xiamen 361028, China. [16] Whole-Genome Research Core Laboratory of Human Diseases, Chang Gung Memorial Hospital, Keelung 204, Taiwan. Correspondence and requests for materials should be addressed to W.-H.C. (email: chung1@cgmh.org.tw) or (email: wenhungchung@yahoo.com) or to S.-I.H. (email: sihung@cgmh.org.tw) or (email: hungshueniu@gmail.com)

Drug hypersensitivity is an important clinical issue, which shows different presentations and pathogenesis[1,2]. T-cell-mediated delayed-type drug hypersensitivity ranges from mild skin rash to life-threatening severe cutaneous adverse reactions (SCAR), including Stevens–Johnson syndrome (SJS), toxic epidermal necrolysis (TEN), and drug reaction with eosinophilia and systemic symptoms (DRESS)[3,4]. Most of SCAR are unpredictable and carry high mortality rates. Some of drug hypersensitivity reactions have shown HLA (human leukocyte antigen) genetic predisposition, e.g., HLA-B*15:02 for carbamazepine (CBZ)-induced SJS/TEN[5], HLA-A*31:01 for CBZ-DRESS[6], HLA-B*58:01 for allopurinol-SCAR[7,8], HLA-B*13:01 for dapsone hypersensitivity[9,10], and HLA-B*57:01 for abacavir hypersensitivity[11,12]. In addition to HLA, the genetic polymorphisms of drug metabolic enzyme CYP2C9 have been linked to phenytoin-induced SCAR[13,14]. However, both the HLA and CYP genetic variants have low positive predictive values (PPV) (e.g., the PPV of HLA-B*15:02 for CBZ-SJS/TEN is only 3%)[1,15], suggesting that other factors are involved in the pathogenesis of SCAR.

T lymphocytes are suggested to play important roles in SCAR, as the cytokine/biomarker signatures reveal the Th1 pathway for DRESS and cytotoxic T lymphocytes (CTL) profile for SJS/TEN[16,17]. Our previous studies discovered that CTL predominantly infiltrated in the skin lesions of SJS/TEN, and expressed inflammatory cytokines and cytotoxic proteins, including granulysin, a key mediator to cause keratinocyte death in SJS/TEN[18,19]. The in vitro lymphocyte activation tests confirm the presence of drug-specific T cells/clones in SCAR[20]. However, it remains unclear how T cells recognize the drug antigens, how the T-cell receptor (TCR) repertoire is used, and whether drug-specific TCR clonotypes mediate the hypersensitivity reactions. In this study, we enroll patients with various drug-induced SCAR (65 SJS/TEN, 8 DRESS) from different ethnic populations, and tolerant/healthy controls. We apply next-generation sequencing (NGS) and single-cell sequencing to investigate TCR repertoire, and further perform functional analyses, molecule modeling, co-cultures, and adoptive cellular transfer of TCR-T to HLA-transgenic mice to elucidate the roles of TCR in the immune synapse of SCAR.

Here, we report the discovery of preferential TCR clonotypes from the blister cells of the skin lesions of SJS/TEN patients. We identify a public TCR composed of a paired TCRα CDR3 (third complementarity-determining region) "VFDNTDKLI" and TCRβ CDR3 "ASSLAGELF" clonotypes from CBZ-SJS/TEN patients recruited from Asia and Europe. This public TCR shows drug-specificity and phenotype-specificity in an HLA-B*15:02-favored manner. Our data of functional assays, co-cultures, and adoptive transfer of TCR-T cells in the mouse model further support that the drug-specific TCR of CTL is essential for the immune synapse that mediates CBZ-SJS/TEN.

## Results
### Clinical demographics of patients and controls. We recruited a total of 73 patients of SCAR, including 8 cases with CBZ-DRESS, and 65 with SJS/TEN caused by carbamazepine (CBZ) ($n = 42$), oxcarbazepine (OXC) ($n = 3$), lamotrigine (LTG) ($n = 4$), phenytoin (PHT) ($n = 6$), and allopurinol (ALP) ($n = 10$). The demographics and HLA genotype data of patients are listed in Supplementary Tables 1 and 2. Among the 42 cases of CBZ-induced SJS/TEN, HLA-B*15:02 allele was found in all 24 (100.00%) of Chinese, 4 of 5 (80.00%) Thai patients, and 2 of 13 (15.38%) subjects enrolled from Europe (Supplementary Table 1). In addition, all three patients with OXC-SJS carried HLA-B*15:02 (Supplementary Table 2). We also enrolled drug-tolerant controls

($n = 12$) who had taken CBZ for more than 6 months without adverse reactions, and six of them carried HLA-B*15:02 (Supplementary Table 3). Furthermore, we recruited 44 healthy donors to represent the general population, who had the phenotype frequency of HLA-B*15:02 as 9% (Supplementary Table 4).

### TCR usage in the blister cells of patients with SJS/TEN. We investigated the TCRβ variable (TRBV) gene usage of the blister cells and peripheral blood mononuclear cells (PBMC) of SJS/TEN patients by NGS, and normalized the expression of TRBV transcripts of each sample to the mean value of the corresponding subtype of healthy donors' PBMC ($n = 44$) (Fig. 1a). Compared with other subtypes, TRBV12-4 exhibited the highest increase of expression (~10- to 100-fold) in the PBMC and blister cells of patients with CBZ-induced SJS/TEN (Fig. 1a; Supplementary Fig. 1). The principal components analysis (PCA) showed the unique pattern of TRBV gene usage in CBZ-SJS/TEN (Fig. 1b). The mean frequency of TRBV12-4 was 31.62% in blister cells ($n = 7$) and 4.95% in PBMC ($n = 11$) of patients with CBZ-SJS/TEN, but only 0.69% in the PBMC of CBZ-tolerant controls ($n = 12$) (Fig. 2a). In addition, the TRBJ2-2 gene highly expressed in the blister cells of CBZ-SJS/TEN patients ($n = 7$) with a mean frequency of 22.15% (Supplementary Fig. 1). We then mapped the V–J junction Circos plot and found the significant increase of TRBV12-4/TRBJ2-2 pairing in the blister cells and PBMC of CBZ-SJS/TEN patients, but not in the CBZ-tolerant controls (Fig. 2b–d; Supplementary Fig. 2).

### Identification of a public TCRβ clonotype from CBZ-SJS/TEN. We investigated the CDR3 assemblies of TCRβ clonotypes in the samples of blister cells of patients with SJS/TEN. Preferential and oligoclonal TCRβ clonotypes were noticed in the blister samples of patients with CBZ-SJS/TEN (Table 1). Remarkably, an abundant TCRβ CDR3 clonotype "ASSLAGELF" was identified in all of the blister samples ($n = 7$, mean: 13.57%, range: 1.08–42.85%) (Table 1 and Fig. 2e). By comparison, this TCRβ clonotype was undetectable or scarce (<0.001%) in the PBMC of CBZ-tolerant controls or healthy donors (Table 1). The expression of TCRβ "ASSLAGELF" clonotype showed drug-specificity, as it was present only in the blister cells of SJS/TEN caused by CBZ, but not other culprit drugs (e.g., phenytoin or allopurinol) (Supplementary Fig. 3). In addition to the public CDR3 "ASSLAGELF", there were similar clonotype clusters with one residue difference (e.g., "ASSLSGELF", "ASSFAGELF", etc.) identified in the blister cells of patients with CBZ-SJS/TEN (Fig. 2f; Supplementary Table 5). The data of flow cytometry further revealed that the specific TRBV12-4 TCR was predominantly expressed by memory CD8+ CTL, which accounted for the majority (72.4%) of the blister cells of CBZ-SJS/TEN (Fig. 2g, h; Supplementary Figs. 4, 5).

### Expression patterns of the public TCRβ. We analyzed the expression of this public TCRβ clonotype "ASSLAGELF" in the PBMC samples of CBZ-SJS/TEN patients enrolled from different ethnic populations. This specific CDR3 clonotype was detected in 0.1–3.6% (mean: 1.34%) of the total TCR reads of NGS data in the PBMC of Han Chinese patients; all had HLA-B*15:02 (Fig. 3a–d). In addition, this specific clonotype was also found in the PBMCs of European patients ($n = 6$; range: 0–4.2%; mean: 1.28%), though most of them had no HLA-B*15:02 allele (Fig. 3b–d). We found that the expression of this public TCRβ clonotype is phenotype-specific, as it was predominately in the PBMC of CBZ-SJS/TEN ($n = 11$; mean: 1.31%), but scarce or absent in CBZ-DRESS ($n = 8$; mean: 0.32%), CBZ-tolerant controls ($n = 12$; mean: <0.001%) or healthy donors ($n = 44$; mean: <0.001%) (Fig. 3c, d).

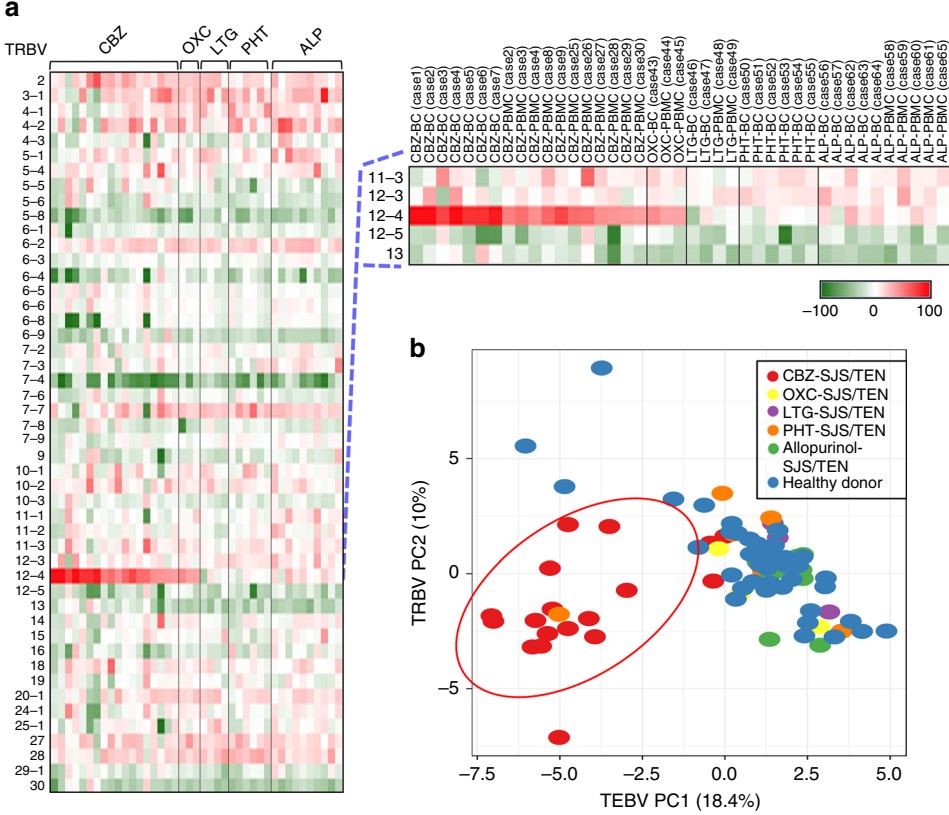

**Fig. 1** Preferential TCR usage in blister cells and PBMC from patients with SJS/TEN. The PBMC and blister cells were isolated from patients with SJS/TEN caused by carbamazepine (CBZ; $n = 18$), oxcarbazepine (OXC; $n = 3$), lamotrigine (LTG; $n = 4$), phenytoin (PHT; $n = 6$), or allopurinol (ALP; $n = 10$). The expression profiles of TCR repertoire in the samples were analyzed by next-generation sequencing. **a** Heatmaps display the expression values of the *TRBV* genes from each sample, which were normalized to the mean values of the corresponding gene of the healthy donors' PBMC ($n = 44$), respectively. The definition of *TRBV* and *TRBJ* was based on the IMGT (ImMunoGeneTics) database. **b** Principal component analysis (PCA) of *TRBV* gene usage across samples from patients ($n = 41$) with SJS/TEN caused by different drugs, and healthy donors ($n = 44$)

Furthermore, we validated the NGS data by quantitative PCR, and confirmed that the public TCRβ expression showed correlation with the disease states of CBZ-SJS/TEN, which had highest levels in blister cells, moderate in the PBMC from the active stage of patients, decrease in the recovery stage, but undetectable in the tolerant controls or healthy donors (Fig. 3e, f; Supplementary Table 6).

**TRVA/TRVB pairing and gene profile of the CTL**. We comprehensively analyzed the TCRα/TCRβ repertoire and the genes expression profile of the corresponding T lymphocytes in the blister cells of patients with CBZ-SJS/TEN by NGS and single-cell sequencing. The pooled data of the blister cells suggested oligoclonal expansion of T lymphocytes with paired TCRα CDR3 "VFDNTDKLI" and TCRβ CDR3 "ASSLAGELF" clonotypes, which accounted for 44.81 and 42.85%, respectively, in a representative data set (Supplementary Fig. 6). To validate the TRVA/TRVB pairing and gene expression profile, we sorted 30 single cells using flow cytometry with monoclonal antibodies (mAb) against TRBV12-3/TRBV12-4. Then the paired TRVA/TRVB, and 18 phenotypic genes were determined by single-cell sequencing. All (100%) of the 30 cells expressed the same specific TCRβ CDR3 clonotype "ASSLAGELF", and 25 (83.33%) cells expressed the TCRα CDR3 clonotype "VFDNTDKLI" (Fig. 4a, b). Two other TCRα CDR3 clonotypes "AASPPDGNQFY" and "ALDIPNFGNEKLT" were found to pair with the specific TCRβ, but presented in low frequencies (13.33 and 3.33%, respectively) in the blister cells (Fig. 4a, b). The expression of the specific TCRα

CDR3 clonotype "VFDNTDKLI" was further validated by Taq-Man quantitative real-time PCR in the blister cells ($n = 5$) or PBMC samples ($n = 6$) of CBZ-SJS/TEN patients, CBZ-DRESS patients ($n = 5$), and controls ($n = 4$) (Supplementary Table 6 and Supplementary Fig. 7). The expression levels of TCRα CDR3 "VFDNTDKLI" and TCRβ CDR3 "ASSLAGELF" showed a similar trend, supporting the correct pairing (Supplementary Fig. 7).

Single-cell sequencing on the 30 T lymphocytes with the public TCRβ "ASSLAGEL" clonotype revealed the abundant transcripts of genes, including *GNLY*, *GZMB*, *IL17A*, *IL12A*, *IL10*, and *RORC* (Fig. 4c). *GNLY* and *GZMB* encode cytotoxic proteins, which are produced by CTL and responsible for disseminated keratinocyte death in SJS/TEN[18]. *IL17A* and *IL12A* are inflammatory cytokines[21,1], and RAR-related orphan receptor C (*RORC*) is important for lymphoid organogenesis and thymopoiesis[22]. Taken together, the public αβTCR was expressed by CD8+ cytotoxic T lymphocytes with abundant cytotoxic proteins (e.g., granulysin and granzyme B), and inflammatory cytokines, including IL17A, IL12A, and RORC.

**The public TCR shows binding affinity toward CBZ**. To investigate the functional role of the public αβTCR, we generated the cDNA construct encoding for a soluble single-chain αβTCR (scTCR) recombinant protein composed of the paired TCRα CDR3 "VFDNTDKLI" and TCRβ CDR3 "ASSLAGELF" (Supplementary Table 7). The scTCR construct was transfected to HEK293F cells, and the recombinant protein was purified from

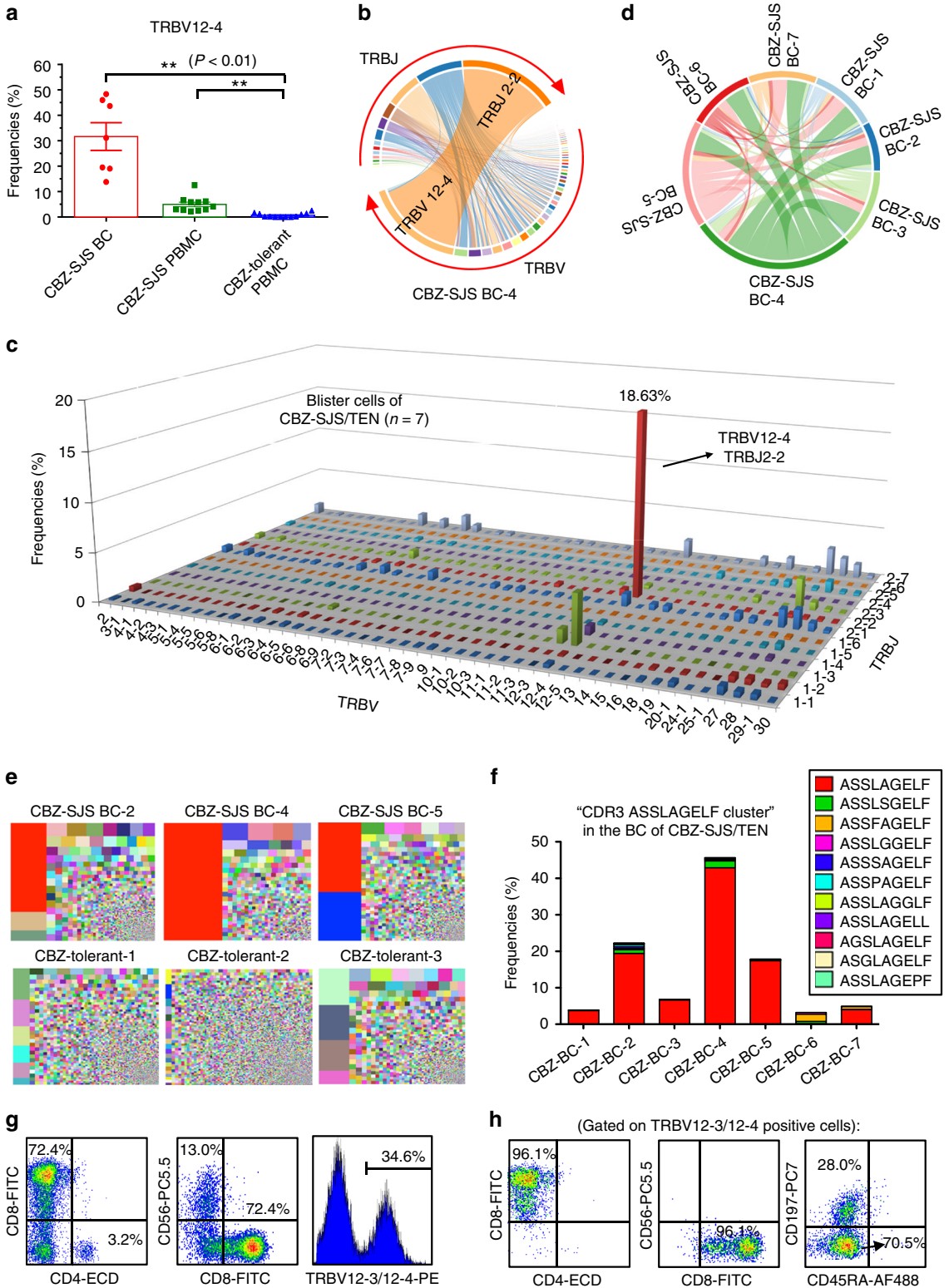

the serum-free cultured medium (see the Methods section and Supplementary Fig. 8). We coated the scTCR recombinant protein on the chip of the BIAcore surface plasmon resonance (SPR) system. Different drugs and metabolites flowed through the chip, and only CBZ, CBZ-10,11-epoxide (CBZ-E), and OXC showed binding response to the public scTCR recombinant protein (Fig. 5; Supplementary Fig. 9). By comparison, the control scTCR protein composed of TCRα CDR3 "AGHDYKLS" and TCRβ

CDR3 "ASTSGPNEQF" did not display binding response to CBZ, CBZ-E or OXC (Supplementary Fig. 10).

We further performed in silico modeling to investigate the potential interaction of the public αβTCR, the drug antigen, and HLA-B*15:02. First, we modeled a peptide complexed to HLA-B*15:02 based on the crystal structure of HLA-B*15:01 (PDB 1XR8) and a peptide motif identified from CBZ-treated cells[23]. We then modeled the public αβTCR interaction with the peptide/

**Fig. 2** Identification of a public TCR from the blister cells of patients with CBZ-SJS/TEN. The TCR repertoire of samples was analyzed by next-generation sequencing. **a** The frequencies of *TRBV*12-4 gene usage were significantly higher in PBMC ($n = 11$) and blister cells ($n = 7$) of patients with CBZ-SJS/TEN, when compared with the PBMC of CBZ-tolerant controls ($n = 12$). The results are expressed as mean ± s.e.m. with each dot representing the data of an individual. Statistical analysis was generated using an unpaired, two-tailed Student's *t* test. **P < 0.01. **b** A representative *TRBV–TRBJ* junction Circos plot of blister cells of a CBZ-SJS/TEN patient (case 4). Arcs correspond to different V and J segments. Ribbons represent V/J pairings with sizes scaled to pairing frequency. **c** The mean frequencies (%) of *TRBV/TRBJ* pairings in the blister cells from CBZ-SJS/TEN patients ($n = 7$). The *x*- and *y*-axis represent the TRBV and TRBJ regions, respectively, and the *z*-axis indicates the mean frequencies of TCRβ rearrangements. **d** The pairwise overlap Circos plot shows the overlapping CDR3 clonotypes among the seven blister samples of cases 1–7. **e** Treemaps of TCRβ CDR3 clonotypes for the representatives of blister cells of patients with CBZ-SJS/TEN (upper panel) and the PBMC of CBZ-tolerant controls (lower panel). Colors represent individual specific CDR3 clonotypes, and the area of each color square represents the frequency in the sample. The specific TCRβ CDR3 "ASSLAGELF" is labeled in red. **f** The frequencies of "CDR3 ASSLAGELF cluster" which includes the TCR clonotypes carrying similar CDR3 sequences with one amino acid difference in the blister cells of CBZ-SJS/TEN patients. **g**, **h** Flow-cytometry analysis of the cell populations expressing the markers of CD4, CD8, CD56, TRBV 12–3/12–4, CD45RA, and CD197 in the blister cells of patients with CBZ-SJS/TEN

| Table 1 Preferential TCRβ CDR3 clonotypes in blister cells from patients with CBZ-SJS/TEN | | | | | | | | | |
|---|---|---|---|---|---|---|---|---|---|
| Case no. | Culprit drug | Symptom | | CDR3 clonotypes | *TRBV* | *TRBJ* | Frequency (%) | Specific CDR3 reads/total reads | Frequency (%) in PBMC of healthy donors[a] |
| Case 1 | CBZ | SJS/TEN | 1 | ASSLSDTIY | 12–4 | 1–3 | 24.45 | 182480/746406 | 0 |
| | | | 2 | ASSLGGAPY | 12–4 | 2–7 | 8.41 | 62776/746406 | 0 |
| | | | 3 | ASSYNPGTGTEEYEQY | 6–1 | 2–7 | 6.50 | 48516/746406 | 0 |
| | | | 4 | ASTHTGELF | 12–4 | 2–2 | 4.60 | 34325/746406 | 0 |
| | | | 5 | ASSLAGELF | 12–4 | 2–2 | 3.73 | 27829/746406 | <0.001 |
| | | | 6 | ASSPRLAGSTDTQY | 27 | 2–3 | 2.11 | 15748/746406 | <0.001 |
| Case 2 | CBZ | SJS | 1 | ASSLAGELF | 12–4 | 2–2 | 19.39 | 153620/792164 | <0.001 |
| | | | 2 | ASSLSGYEQY | 27 | 2–7 | 4.01 | 31781/792164 | 0.0026 |
| | | | 3 | ASSSAGEVF | 12–4 | 2–1 | 2.27 | 17990/792164 | 0 |
| Case 3 | CBZ | SJS | 1 | ASSLAGELF | 12–4 | 2–2 | 6.64 | 5411/81449 | <0.001 |
| | | | 2 | ATSGPNQETQY | 24–1 | 2–5 | 2.08 | 1697/81449 | 0 |
| Case 4 | CBZ | TEN | 1 | ASSLAGELF | 12–4 | 2–2 | 42.85 | 31148/72692 | <0.001 |
| | | | 2 | ASSYRDGYEQY | 6–5 | 2–7 | 2.73 | 1986/72692 | <0.001 |
| | | | 3 | ASSRRAVVGSYNEQF | 7–2 | 2–1 | 2.62 | 1905/72692 | 0 |
| | | | 4 | ATHGTGYLEQY | 25–1 | 2–7 | 2.15 | 1565/72692 | 0 |
| Case 5 | CBZ | SJS/TEN | 1 | ASSLAGELF | 12–4 | 2–2 | 17.41 | 143248/822649 | <0.001 |
| | | | 2 | ASSSRLAGGTDTQY | 27 | 2–3 | 12.11 | 99586/822649 | 0.0025 |
| Case 6 | CBZ | SJS/TEN | 1 | ASSPSDRSSYEQY | 18 | 2–7 | 4.69 | 57694/1229012 | 0 |
| | | | 2 | ASSSLTSSWVEQF | 28 | 2–1 | 3.81 | 46784/1229012 | 0 |
| | | | 3 | ASTSGPNEQF | 12–4 | 2–1 | 2.99 | 36759/1229012 | <0.001 |
| | | | 4 | ASSYSSTDTQY | 6–5 | 2–3 | 2.80 | 34429/1229012 | 0.0029 |
| | | | 5 | ASSQYRYNEQF | 14 | 2–1 | 2.60 | 32006/1229012 | 0 |
| | | | 6 | ASSFAGELF | 12–4 | 2–2 | 2.22 | 27338/1229012 | <0.001 |
| | | | 7 | ASSLAGELF | 12–4 | 2–2 | 1.08 | 13273/1229012 | <0.001 |
| Case 7 | CBZ | TEN | 1 | ASSWDPTIY | 12–3 | 1–3 | 10.21 | 81449/798017 | 0 |
| | | | 2 | ASSLAGELF | 12–4 | 2–2 | 4.00 | 31881/798017 | <0.001 |
| | | | 3 | ASIDGSSLNEQF | 27 | 2–1 | 3.57 | 28498/798017 | 0 |
| | | | 4 | ASSLSGYEQY | 27 | 2–7 | 2.90 | 23122/798017 | 0.0026 |
| | | | 5 | ASSYSDTIY | 6–6 | 1–3 | 2.50 | 19948/798017 | 0 |

*CDR3* third complementarity-determining region, *PBMC* peripheral blood mononuclear cell, *SJS* Stevens–Johnson syndrome, *TEN* toxic epidermal necrolysis, *TRBJ* T-cell receptor β joining, *TRBV* T-cell receptor β variable
[a]The mean frequencies of the corresponding CDR3 clonotypes in PBMC of healthy donors ($n = 44$)

HLA-B*15:02 complex in a conventional docking orientation[24]. Molecular docking was applied to predict CBZ binding to the HLA-B*15:02/peptide/TCR complex[25]. Molecular docking suggests that CBZ is more likely to bind the solvent exposed portion of the interface between the public αβTCR and HLA-B*15:02 compared with other sites, such as those within the antigen-binding cleft. CBZ was predicted to bind this TCR/HLA interface site with an estimated ΔG -7.9 kcal mol$^{-1}$, which was comprised the α1 helix of HLA-B*15:02, TCRα CDR3 VFDNTDKLI, and TCR β CDR2 (Supplementary Fig. 11).

**Public TCR mediates immune response to the drug and HLA.** To investigate the functional role of the public αβTCR in the immune synapse, we generated the C1R-HLA-B*15:02 transfectants (C1R-B*1502) as the antigen presenting cells (APC) and the αβTCR transfectants (5KC-TCR) expressing the public TCRα CDR3 and TCRβ CDR3 (Supplementary Table 8). Incubation of 5KC cells with CBZ and C1R-HLA-B*1502 showed no reactivity (Fig. 6a, b). By comparison, the cultures of 5KC-TCR and C1R displayed increased IL-2 production upon CBZ drug stimulation (Fig. 6a, b). The strongest IL-2 enzyme-linked immunospot (ELISPOT) response was observed in the co-cultures of 5KC-TCR transfectants, C1R-B*15:02 cells, and CBZ (Fig. 6a, b). In addition to CBZ, 5KC-TCR transfectants also reacted to CBZ-10,11-epoxide and OXC (Fig. 6c). CBZ of 100 μmol L$^{-1}$ or CBZ-10,11-epoxide of 50 μmol L$^{-1}$ stimulated the maximal number of 5KC-TCR cells producing IL-2 (Fig. 6d). These results suggested that

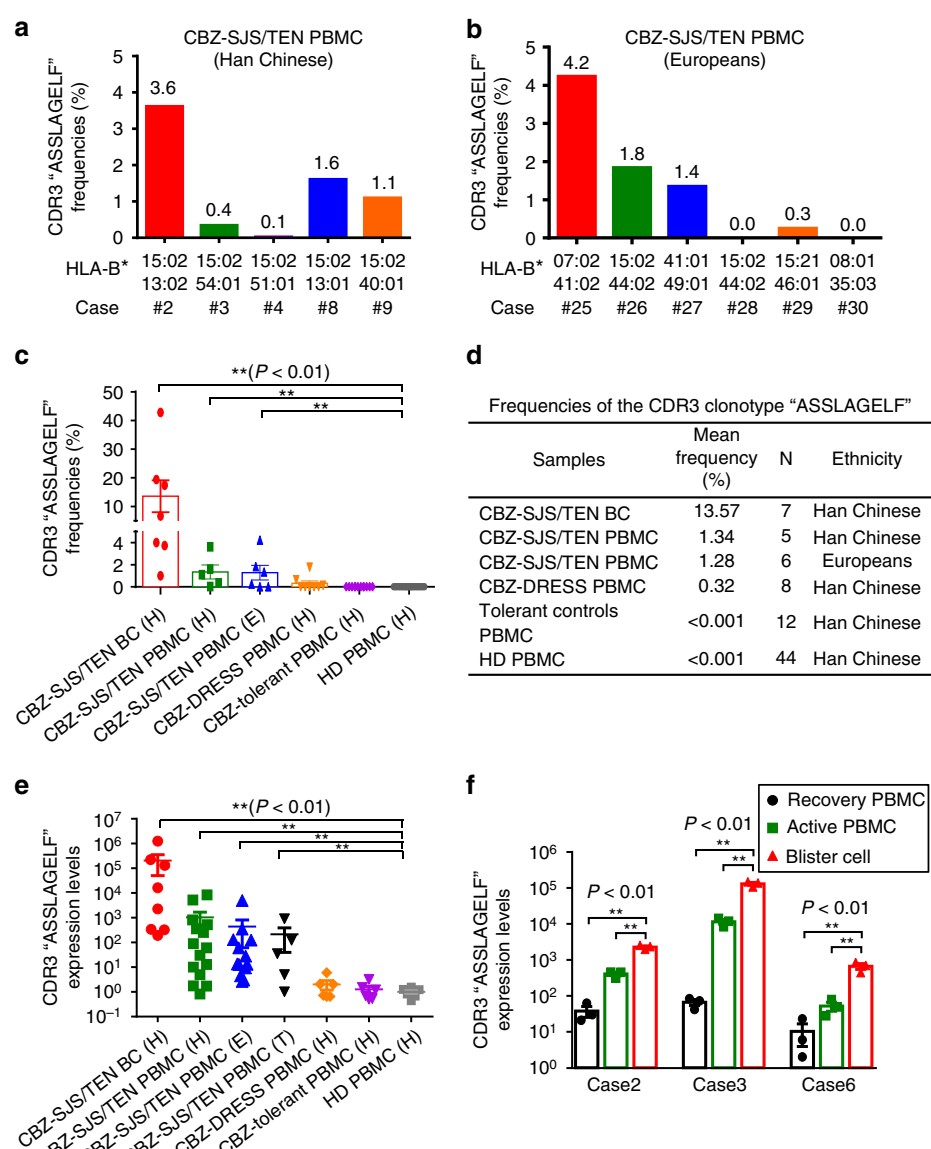

**Fig. 3** Expression of the public TCRβ CDR3 clonotype in the CBZ-SCAR patients. **a**, **b** The frequencies of the specific CDR3 "ASSLAGELF" were determined by TCR NGS using the PBMC of patients of Han Chinese ($n = 5$) and Europeans ($n = 6$) with CBZ-SJS/TEN. **c**, **d** Comparison of the frequencies of the CDR3 "ASSLAGELF" in the blister cells (BC) and PBMC of patients with CBZ-SCAR, CBZ-tolerant controls, and healthy donors (HD). The results are expressed as mean ± s.e.m. with each dot representing the data of an individual. Statistical analysis was generated using an unpaired, two-tailed Student's $t$ test. **e** Determination of the expression levels of the specific CDR3 by quantitative real-time PCR in the samples of BC and PBMC of patients with CBZ-SCAR of different ethnic populations including Han Chinese (H), Thai (T), and Europeans (E). The samples used for analysis include blister cells of CBZ-SJS/TEN (Han Chinese ($n = 8$)), PBMC of CBZ-SJS/TEN (Han Chinese ($n = 15$; Europeans ($n = 13$); Thai people ($n = 5$)), PBMC of CBZ-DRESS (Han Chinese ($n = 6$)), PBMC of tolerant controls (Han Chinese ($n = 6$)), and PBMC of healthy donors (Han Chinese ($n = 6$)). The expression level of the specific TCRβ clonotype was normalized by that of CD3, and the detection limit of the TCRβ clonotype/CD3 ratio was 0.0001. The results are expressed as mean ± s.e.m. with each dot representing the data of an individual. Statistical analysis was performed using an unpaired, two-tailed Student's $t$ test. **f** Determination of the expression levels of the specific CDR3 by quantitative real-time PCR in the samples of BC and PBMC of patients with CBZ-SCAR in the active or recovery disease states. The results are representative data of three cases and each with triplicate measurements. The data are expressed as mean ± s.e.m. with each dot representing the data of one sample. Statistical analysis was performed using an unpaired, two-tailed Student's $t$ test. **$P < 0.01$

the public αβTCR reacts to CBZ and its structural analogs, and the HLA-B*15:02 presenting promotes the immune recognition.

**Adoptive transfer of TCR-T cells to *HLA-B*15:02* transgenic mice.** We generated *HLA-B*15:02* transgenic mice and confirmed the stable expression of transgene (Supplementary Fig. 12). The transgenic mice were assigned to three groups: (I) the vehicle

controls; (II) oral administration of carbamazepine (328 mg kg$^{-1}$ per day); (III) given carbamazepine and adoptive transfer of the public αβTCR-T lymphocytes. In the group II, the transgenic mice had received carbamazepine for more than 3 months; however, no phenotypes of SCAR developed. By comparison, the HLA-B*15:02$^+$αβTCR$^+$CBZ$^+$ mice (group III) displayed phenotypes mimicking SCAR with multi-organ injuries after 4 weeks of adoptive cell transfer (Fig. 7). The group III transgenic mice

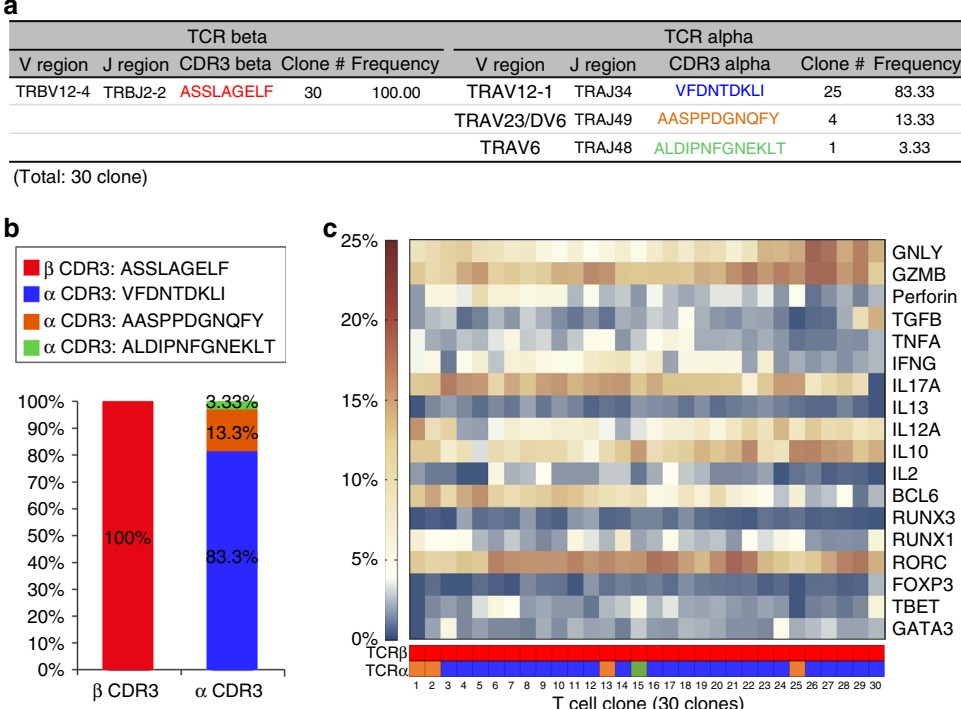

**Fig. 4** Single-cell sequencing on the TCRα/TCRβ gene pairing and gene profile. The blister cells of a patient with CBZ-SJS/TEN (case 4) were sorted by flow cytometry using mAb recognizing human TCR TRBV12-3/TRBV12-4 (Beckman Coulter). A total of 30 lymphocytes were analyzed by single-cell sequencing to determine the V/J usage, CDR3 TCRα/TCRβ clonotypes, and the transcripts of 18 genes. **a**, **b** The specific TCRβ CDR3 clonotype "ASSLAGELF" was detected in all (100%, 30/30) single-cell clones. Twenty-five (83.33%) clones expressed the TCRα CDR3 clonotype "VFDNTDKLI", four (13.33) clones expressed TCRα CDR3 "AASPPDGNQFY", and one (3.33%) clone expressed TCRα CDR3 "ALDIPNFGNEKLT". **c** A heatmap displays the expression values of 18 genes related to the function and differentiation of T lymphocytes in 30 single-cell clones. The corresponding TCRα and TCRβ CDR3 clonotypes are indicated to each clone

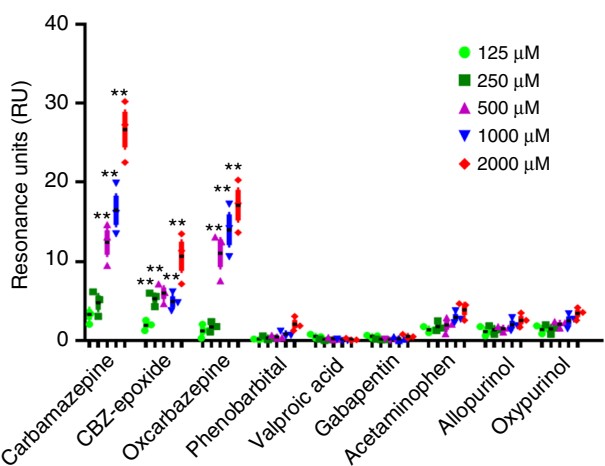

**Fig. 5** Binding response of the public TCR to different drugs and structural analogs. Surface plasmon resonance (SPR) was applied for analyzing the binding response between the soluble single-chain αβTCR (scTCR) recombinant protein and drug compounds. The scTCR composed of the TCRα CDR3 "VFDNTDKLI" and TCRβ CDR3 "ASSLAGELF" was purified from the cultured medium of HEK293F transfectants. The scTCR protein was coated to the chip, and different drugs or metabolites flowed through the chip. The binding response of the scTCR protein toward CBZ and related compounds was examined by SPR. The results are representative of three independent experiments, and expressed as mean ± s.e.m. with each dot representing the data of one sample. Statistical analysis was generated using an unpaired, two-tailed Student's $t$ test. **$P < 0.01$

showed hair loss, perioral/paranasal mucositis, skin erythema with telangiectasia, and conjunctivitis (Fig. 7a). Histology and immunohistochemistry of the biopsies of affected skin showed marked dermal inflammatory cell infiltration with epidermal dyskeratosis (Supplementary Fig. 13), and elevated levels of cytotoxic proteins and inflammatory cytokine (e.g. granzyme B, IFNγ, and TNFα) (Fig. 7b–e). In addition, there were increased amounts of IFNγ + CD4 + or IFNγ + CD8 + T cells in the peripheral blood (Fig. 7f–j). Furthermore, compared with the controls, the HLA-B*15:02⁺αβTCR⁺CBZ⁺ mice (group III) displayed the impaired liver and kidney function with elevated serum ALT/GPT, BUN, and CRE levels (Fig. 7k–m). Taken together, these data support the essential role of the public αβTCR in the formation of an immune synapse that mediates SCAR.

## Discussion

Different genetic and nongenetic factors predispose individuals to SCAR. The germline *HLA* alleles and *CYP* variants have been linked to SCAR, and some of the genetic markers have been translated to clinical applications to prevent the drug hypersensitivity reactions[8,12,14]. The functional studies further demonstrated that the associated HLA alleles possess increased affinity to the culprit drugs/metabolites[26,27] and drug-specific T cells serve the immunological basis of SCAR[18,28,29]. The genetic defects on drug metabolism enzymes or impaired renal function in SCAR patients caused increase of drug allergen and induced lymphocyte activation and hypersensitivity reactions[13,30]. However, most of SCAR do not show a strong genetic predisposition in the germline HLA or drug metabolism genes. Herein, we

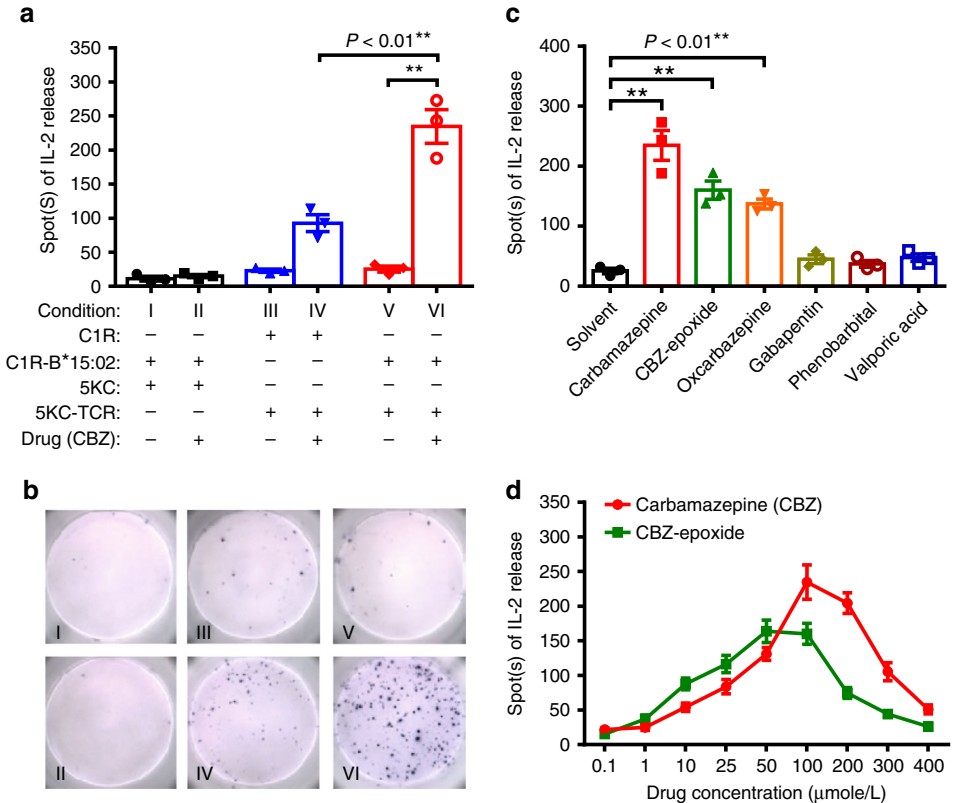

**Fig. 6** Immune response of the public αβTCR transfectants to the drug antigen and HLA. **a** Parental 5KC or 5KC-TCR cells were co-cultured with C1R or C1R-B*15:02 cells and stimulated with CBZ (25 μg/ml) or solvent control. The immune response of the 5KC-TCR transfectants was determined by the IL2 ELISPOT assay. The results are representative of three independent experiments, and expressed as mean ± s.e.m. with each dot representing the data of one sample. Statistical analysis was performed using an unpaired, two-tailed Student's *t* test. **b** A representative IL2 ELISPOT data set of Fig. 6a is shown. **c** Response of 5KC-TCR and C1R-B*15:02 co-cultures to different antiepileptic drugs at physiological concentrations (i.e., CBZ: 25 μg ml$^{-1}$; CBZ-epoxide: 10 μg m$^{-1}$; oxcarbazepine: 25 μg ml$^{-1}$; gabapentin: 6 μg ml$^{-1}$; phenobarbital: 15 μg ml$^{-1}$; valproic acid: 100 μg ml$^{-1}$). The results are representative of three independent experiments, and expressed as mean ± s.e.m. with each dot representing the data of one sample. Statistical analysis was performed using an unpaired, two-tailed Student's *t* test. **d** Responses of 5KC-TCR and C1R-B*15:02 co-cultures to CBZ or CBZ-epoxide at serial concentrations (in micromoles). The data are shown as mean ± s.e.m. of triplicate measurements for each condition and each dot representing the data of one sample. The data were analyzed using an unpaired, two-tailed Student's *t* test. "5KC-TCR" represents 5KC cells expressing the public αβTCR composed of the paired TCRα CDR3 "VFDNTDKLI" and TCRβ CDR3 "ASSLAGELF" clonotypes. CBZ carbamazepine. **$P < 0.01$

investigate the immune repertoire of SCAR, and evaluate the role of TCR in the pathogenesis of SCAR.

The development of NGS advances the understandings of TCR repertoire. This study applies NGS and uncovers an abundant, public, and unique TCR clonotype from SCAR. The TCR clonotype identified in this study is different from our previous report, which utilized the traditional cloning and Sanger sequencing method with the samples of co-cultures of EBV-transformed B cells as the APC and the PBMC of CBZ-SJS/TEN patients[31]. The in vitro expansion of T cells by co-culturing with EBV-transformed B cells as APC may distort the T-cell repertoire[32]. To reduce the methodological biases, we here use the blister cells of SJS/TEN patients enrolled from a large cohort, and apply multiplex PCR of TCR subtype-specific primers (i.e., iRepertoire® library preparation system) for NGS and single-cell sequencing, and validate the TCR function by transfection and adoptive T-cell transfer into the *HLA-B*15:02*-transgenic mice. The iRepertoire®-based PCR library preparation system and single-cell sequencing method used in this study have been applied by many studies described in the literature[33–36]. Rosati et al. compared the methods of different library preparation systems, including iRepertoire®-based PCR, 5′ RACE-based PCR with or without UMI correction, and showed that the percentages

of sequences representing abundant clonotypes captured by these three methods are similar[37]. Following NGS, our quantitative real-time PCR and functional assays further proved that the public TCR clonotype is essential for the formation of the immune synapse of CBZ-SJS/TEN.

To explain the interaction of HLA, drug antigen, and TCR in drug hypersensitivity, there are different hypotheses, including the "hapten" theory[38], the "pharmacological interaction with immune receptors (p-i)" concept[2], the "altered peptide repertoire" model[23,39], and the "altered TCR repertoire" model[40]. We previously applied mass spectrometry to evaluate the peptide repertoire of HLA-B*1502, but found no evidence suggests the presence of CBZ haptenated peptide[41]. Our previous studies further demonstrated that the chemical antigens (e.g., CBZ or oxypurinol) could directly interact with HLA proteins without the involvement of the antigen-processing pathway, which supports the "p-i" model[42–44]. In this study, we identified a predominate and public αβTCR clonotype, which can directly bind to CBZ and its structural analogs (e.g., CBZ-10,11-epoxide and OXC), and the immune response was promoted by the presence of HLA-B*15:02. The oligoclonal TCR clonotype identified in CBZ-SJS/TEN further supports the "p-i" concept, but not the "altered peptide repertoire" model, which induces

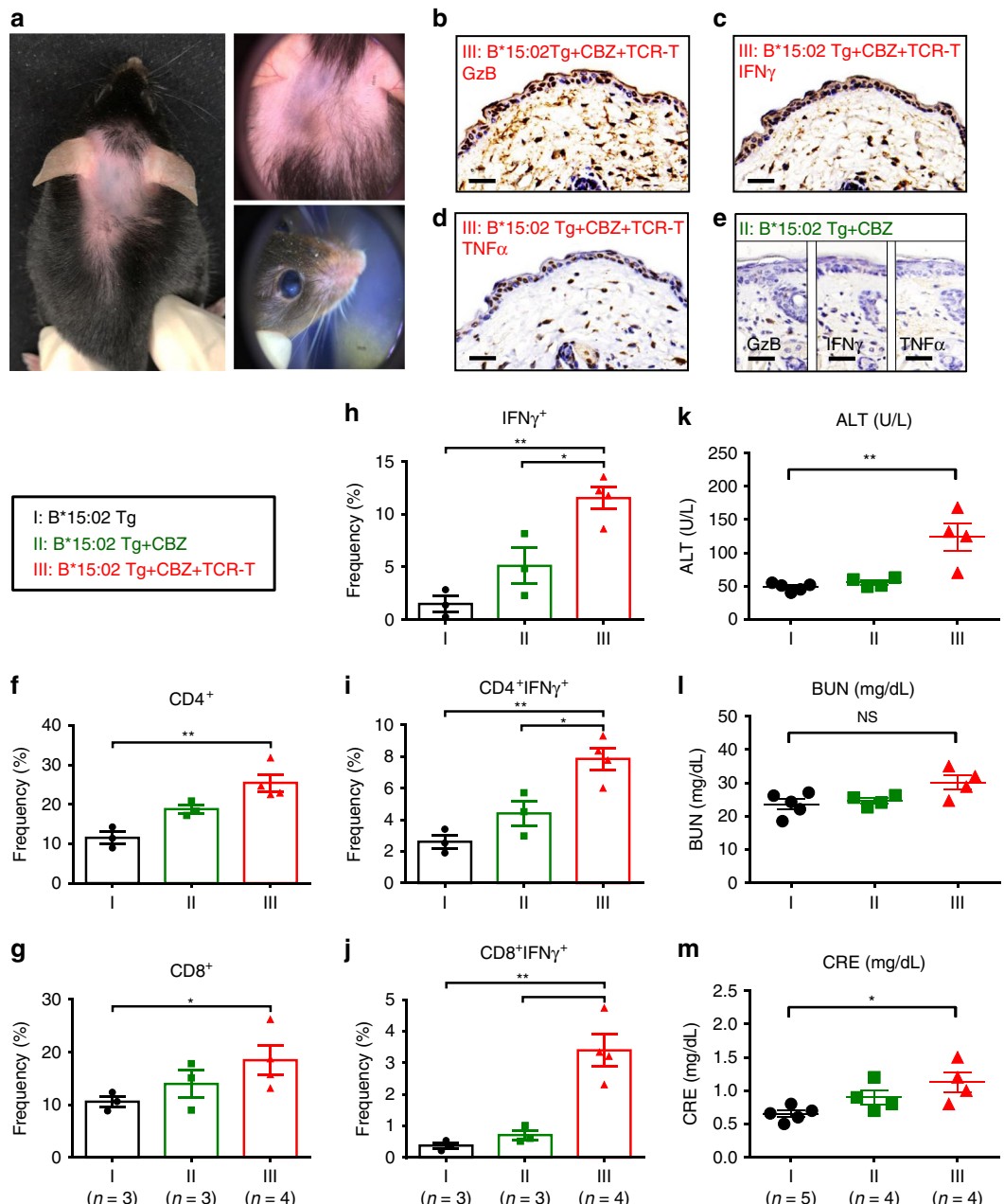

**Fig. 7** Adoptive transfer of TCR-T lymphocytes to HLA-B*15:02 transgenic mice. HLA-B*15:02 transgenic mice were assigned to the group (I) the vehicle controls (*n* = 5); group (II) given carbamazepine daily (328 mg/kg/day) by oral gavage (*n* = 4); and group (III) given carbamazepine daily and adoptive transfer of the public TCR-T (*n* = 4). Photographs, biopsies of the affected skin, and peripheral blood were obtained. **a** Representative photos of the affected skin and eyes of the group III mice are shown. **b–e** Immunohistochemistry staining of the skin biopsies of the mice. Compared with the group II, the group III mice showed augmented expression of cytotoxic protein granzyme B (GzB), and inflammatory cytokines, including IFNγ and TNFα. The results are representative of three independent experiments. Scale bar indicates 100 μm. **f–j** The frequencies of CD4+, CD8+, and/or IFN+ lymphocytes in the peripheral blood of the mice group I (*n* = 3), group II (*n* = 3), and group III (*n* = 4). The results are expressed as mean ± s.e.m. with each dot representing the data of an individual mouse. Statistical analysis was performed using an unpaired, two-tailed Student's *t* test. **k–m** The plasma levels of ALT (alanine aminotransferase), BUN (blood urea nitrogen), and CRE (creatinine) were determined in the mice group I (*n* = 5), group II (*n* = 4), and group III (*n* = 4). The levels of each parameter are plotted as the mean ± s.e.m. with each dot representing the data of an individual mouse. Statistical analysis was performed using an unpaired, two-tailed Student's *t* test. *$P < 0.05$; **$P < 0.01$

polyclonal TCR[23,39]. These T lymphocytes with a public TCR recognizing a small chemical antigen presented by the preferred HLA molecule, may arise from the preexisting memory T cells by heterologous immune response[4]. Whether the public TCR could be generalized to SJS/TEN caused by other drugs needs further studies.

Different animal models for drug hypersensitivity have been reported. Saito et al. transplanted the PBMCs and skin tissue of patients with SJS/TEN to mice, and the mice showed marked conjunctival congestion and dead keratinocytes in the darkening skin-grafted areas upon receiving the causative drug[45]. However, xenografts induced immune rejection could not be excluded in

the study. Recently, Cardone et al. generated *HLA-B\*57:01*-transgenic mice and showed that the drug was tolerated in vivo; depletion of CD4 + T cells prior to abacavir administration was required to induce the reactive CD8 + T cells infiltration and inflammation in the drug-sensitized skin of mice[46]. In this study, we generated an animal model of SCAR. The adoptive transfer of the public TCR-T and oral administration of CBZ led to the development of symptoms of SCAR in the *HLA-B\*15:02*-transgenic mice. The method of TCR-T adoptive cell transfer has been applied in many studies, especially in the field of cancer immunotherapy targeting neoantigens[47–50]. The absence of blistering skin reactions in our animal model may relate to the fact that mice lack granulysin (*GNLY*) gene, which mediates the extensive epidermal necrosis in SJS/TEN[18]. Our animal studies demonstrate the essential role of TCR in the pathogenesis of SCAR.

In summary, we identify the preferential TCR usage in patients with SCAR. A public TCR composed of a paired TCRα CDR3 "VFDNTDKLI" and TCRβ CDR3 "ASSLAGELF" clonotypes and its similar TCR clusters are found in the blister cells and peripheral blood of CBZ-SJS/TEN patients enrolled from different ethnic populations. The public αβTCR is expressed by CTL that also possess abundant cytotoxic proteins and inflammatory cytokines. Our in vitro functional analyses and co-culture experiments show that the public αβTCR on T cells triggers the immune response against drug antigen presented by HLA-B\*15:02. Our animal study further reveals that HLA-B\*15:02-transgenic mice received adoptive transfer of the public TCR-T cells and oral administration of CBZ develop symptoms mimicking SCAR in humans. In conclusion, our results not only support the essential role of this public TCR in the formation of immune synapse that mediates SCAR but also have great potential for clinical applications and development of therapeutics for the disease.

## Methods

**Patients and samples**. We enrolled a total of 129 subjects, including 73 patients with SCAR (65 with SJS/TEN and 8 with DRESS) (Supplementary Tables 1, 2), 12 tolerant controls (Supplementary Table 3), and 44 healthy donors of Han Chinese origin (Supplementary Table 4). Among 73 patients with SJS/TEN or DRESS, there were 55 Chinese enrolled from the Chang Gung Memorial Hospital Health System and Taiwan severe cutaneous adverse reactions (T-SCAR) consortium (including National Taiwan University Hospital, Taichung Veterans General Hospital, National Cheng Kung University Hospital, and Kaohsiung Medical University, and Chung-Ho Memorial Hospital) in Taiwan, 5 from Thailand, and 13 recruited by the RegiSCAR group from Europe, during 2011 and 2018. The diagnosis of SJS/TEN or DRESS was based on the definition of the RegiSCAR study group[51–53]. Only patients with a probable or definite diagnosis of SJS/TEN, or DRESS, were enrolled in this study. The Naranjo algorithm[54] and the algorithm of drug causality assessment for SJS/TEN (ALDEN)[55] were applied to identify the offending drug. In addition, we also enrolled drug-tolerant subjects of Han Chinese (*n* = 12), who had received the drug for more than 6 months without adverse reactions. We collected clinical information and biological samples, including skin biopsies, blister fluids/cells, and the PBMC from patients in the acute or recovery stage. The *HLA-B* genotypes of enrolled subjects were determined by SeCore® HLA Sequence-based typing (Invitrogen, Life Technologies, USA). This study complied with all relevant ethical regulations for work with human participants, and approval for the study was obtained from the institutional review board of the study sites (103-2562C, 104-2664A3, 201601761BO, YM106026F-1, IRB00001189, R.1235-9). Informed consent was obtained from each participant.

**TCR repertoire analysis and single-cell sequencing**. We applied PCR amplification and high-throughput NGS for the VDJ junction and the rearranged CDR3s of TCR[36,56,57]. Briefly, ~100–500 ng of RNA per sample was isolated from blister cells of the skin lesions or from PBMC of the enrolled subjects. The cDNA library of TCR beta chain was produced by multiple PCR using a panel of TCR primers specifically targeting to the V, D, and J gene regions, and the amplicons were sequenced by Illumina Miseq. Single cells from blister samples were sorted by flow cytometry, and the reads and sequences of TCRα/TCRβ and 18 functional genes with granulysin (*GNLY*) in the panel were determined[36]. A series of quality control procedures were established to exclude the false assignments of samples, and eliminate the low-quality sequencing reads[36,56,57]. The CDR3 interval of TCR transcripts was identified as comprising all the amino acids between the Y[YFLI]C

at the 3′ end of the V gene segment and [FW]GXGT (X represents any amino acid) within the J segments[56]. The *TRBV* (T-cell receptor beta variable genes), *TRBJ* (T-cell receptor beta joining genes), and CDR3 clonotypes were defined according to the ImMunoGeneTics information (IMGT) database (www.imgt.org)[36,56–58].

**Flow cytometry**. Flow cytometry was carried out using distinct fluorochrome-conjugated mAb that recognize human CD4, CD8, CD56 (Beckman Coulter), CD45RA, CD197 (BioLegend), human TRBV12-3/TRBV12-4 (Beckman Coulter), and mouse CD4 and CD8 (eBioscience). These mAbs were labeled with Alexa Fluor 488, phycoerythrin (PE), phycoerythrin-Texas Red (ECD), phycoerythrin-cyanin 5 (PC5), or phycoerythrin-cyanin 7 (PC7). The cells were examined by means of multicolor flow cytometry on the Cytomics FC500 flow cytometer (Beckman Coulter), and data were analyzed with CXP software (Beckman Coulter).

**Quantitative real-time PCR**. We isolated the total RNA and obtained cDNA of the PBMC or blister cells by reverse transcription. We quantified the amount of the specific TCRβ CDR3 "ASSLAGELF" of the cDNA samples by Taqman real-time PCR (forward primer: 5′-TTCTCAGCTAAGATGCCTAATGCA-3′, reverse primer: 5′-AAACAGCTCCCCGGCTAAA-3′, probe: 5′-TGAAGATCCAGCCCTC-3′) (Life Technology). The Taqman real-time PCR assay for detecting the TCRα clonotype "VFDNTDKLI" was designed as the forward primer: 5′-CTCAGT-GATTCAGCCACCTACCT-3′, reverse primer: 5′-TGGTCCCAGTCCCAAA-GATG-3′, and probe: 5′-TCGATAACACCGACAAGC-3′ (Life Technology). The expression level of the specific TCR clonotypes was normalized by that of CD3, and the detection limit of the TCR clonotype/CD3 ratio was 0.0001. The number of cycles necessary to reach threshold fluorescence for each gene or β-actin control reaction was calculated at the crossing point (cycle threshold), and the cycle threshold of CD3 or β-actin in each reaction was used as the internal control in parallel experiments.

**Generation of single-chain TCRα/TCRβ recombinant protein**. We generated single-chain TCRα/TCRβ expression constructs (scTCR)[59]. The cDNA of TCRα and TCRβ clonotyps were cloned from the RNA samples of the blister cell from the SJS/TEN patients. The cDNA fragments of TCRα and TCRβ connected by a linker, and then attached to a human antibody Fc region. The single-chain TCRα-linker-TCRβ-Fc insert was cloned into a pcDNA vector (pcDNA/scTCR-Fc) (Supplementary Table 7). The scTCR-Fc plasmid was transfected into the HEK293F cells (Thermo Fisher, Cat: R79007), and the soluble single-chain scTCR-Fc recombinant protein was purified from the culture medium by protein A beads[60].

**Surface plasmon resonance analysis**. A Biacore T200 surface plasmon resonance (SPR) biosensor (GE Healthcare, Piscataway, NJ) was used to analyze the interaction between the scTCR recombinant protein and drugs. For SPR assay, we immobilized the anti-human IgG (Fc) antibody (AP113, Millipore) on sensor chips using an amine-coupling reaction at a density of 10,000 response units. The scTCF-Fc protein flowed through the channel and bound to immobilized antihuman IgG (Fc) antibody, with a binding signal of ~3000 response units. Drugs dissolved in PBS or 5% DMSO/PBS were used, and response of the interaction was reference subtracted and corrected with a standard curve to compensate for solvent effects. The data were analyzed using BIA Evaluation Version 3.1 (GE Healthcare).

**Modeling TCR/peptide/HLA and CBZ**. Homology models of HLA-B\*15:02 and a TCR clonotype with CBZ-specific TCRα CDR3 "VFDNTDKLI" and TCRβ CDR3 "ASSLAGELF" were generated using the SWISS-MODEL workspace[61]. HLA-B\*15:02 presenting the peptide HLASSGHSY was superimposed on HLA-B\*57:01 (PDB 3UPR)[39]. HLASSGHSY was selected for modeling because of similarity to a published motif for peptides eluted from HLA-B\*15:02 in the presence of carbamazepine in drug-treated cells[23]. A TCR in the conventional docking orientation, crystallized in complex with HLA-B\*27:05 and peptide (PDB 4G8G)[24], was positioned by aligning the structure to modeled HLA-B\*15:02. Homology modeled αβTCR chains were superimposed onto the TCR reference structure. Superposition was conducted with COOT[62] using the Secondary-Structure Matching (SSM) program, and subsequent geometry/energy minimization performed using PHENIX[63]. Molecular docking was conducted with AutoDock Vina[25] based on drug-binding sites identified by F pocket[64,65]. Images were generated with PyMOL (PyMOL Molecular Graphics System, Version 1.2, Schrodinger LLC, New York, NY).

**Co-cultures of HLA transfectants and TCR hybridomas**. C1R is a HLA class I-deficient lymphoblastoid cell line (ATCC, CRL-2371™), and we have generated the C1R-HLA-B\*15:02 stable clone APC[41]. The murine 5KC hybridoma lacking TCRα and TCRβ chains was used to reconstitute TCR transfectants[66,67]. Briefly, the full-length cDNA fragments containing the TCRα CDR3 "VFDNTDKLI" and the TCRβ CDR3 "ASSLAGELF" were obtained from the blister cells of SJS/TEN patients. The unique TCR α and β chains were linked to the mouse TCR constant domain by the PTV1.2A sequence (Supplementary Table 8). Then, the cDNA were cloned into MSCV-based retroviral vectors carrying green fluorescent protein (GFP) (pMIGII), followed by production of replication-incompetent retroviruses

encoding TCR sequences[68]. Phoenix cells were co-transfected with pMIGII plasmids and the pCL-Eco packaging vector to produce replication-incompetent retrovirus encoding TCR sequences[69]. The transduced 5KC hybridomas expressing the specific αβTCR CDR3 (5KC-TCR) were used for the antigen specificity assay. We co-cultured the 5KC-TCR transfectants ($1 \times 10^5$ cells) and C1R/C1R-HLA-B*15:02 cells ($1 \times 10^5$ cells) with drugs (25 μg/mL). After 48-h incubation, we measured mouse IL-2 production using a direct cytokine ELISPOT assay (Mabtech). Plates were scanned and analyzed using an ImmunoSpot reader (CTL Cellular Technology).

**Generation of HLA-transgenic mice and adoptive cell transfer**. This study complied with all relevant ethical regulations for animal testing and research, and Experimental Animal Ethics Committee of the institute (National Yang-Ming University) approved the animal protocols of this study (IACUC no.: 1031232; 1041238). This investigation conformed to the US National Institute of Health (NIH) guidelines for the care and use of laboratory animals (Publication no. 85–23, revised 1996). We generated the mono-chain homozygous *HLA-B*15:02* transgenic mice in the C57BL/6 genetic background with triple-knockout of the mouse *MHC* class I genes *H-2b* and *H-2d*, and β₂-microglobulin genes[70]. The stable expression of the *HLA-B*15:02* protein in the transgenic mice was confirmed (Supplementary Fig. 12). The TCR-T lymphocytes were generated by transducing the specific αβTCR construct to the splenocytes of donor mice using the ViraPowerᵀᴹ Lentiviral Expression System (Invitrogen). Adoptive cell transfer was performed by intravenous injection of the CTL isolated from the splenocytes expressing the human public αβTCR into the recipient mice with a dose of $1 \times 10^6$ cells. We divided the HLA-B*15:02 transgenic mice into three groups: (I) vehicle controls, (II) given carbamazepine daily (328 mg kg⁻¹ per day) (Tegretol tablets, Novartis) by oral gavage, and (III) received both carbamazepine and adoptive transfer of the public αβTCR-transfected T lymphocytes (TCR-T) via intravenous injection. The affected skin and eyes were evaluated by dermoscopy using a DermLite 3 Gen dermatoscope in polarized mode at ×10 magnification. Biopsies of the skin and peripheral blood were obtained for immunohistochemistry, flow cytometry, and biochemistry analyses. The mouse serum ALT, BUN, and CRE levels were determined using a chemistry analyzer (FUJI, DRI-CHEM 4000i).

**Histopathological and immunohistochemical staining**. We performed H&E staining and immunohistochemical analyses using the paraffin sections of mouse skin biopsies with mAb against mouse granzyme B (clone TA312131, OriGene), IFNγ (clone bs-0480R, Bioss), or TNFα (clone ab6671, Abcam). The secondary antibodies conjugated to peroxidase and the DAB Detection Kit (Dako) were used for the following staining. The control slides were incubated with the secondary antibody, or isotype control antibodies alone.

**Statistical analysis**. Significant differences between the groups were analyzed using an unpaired, two-tailed Student's *t* test. The heatmaps were generated using the built-in R heatmap() function in stats package, the Circos plots by VDJtools software (MiLaboratory), and the treemaps by the treemap() function version 1.2.0.1 (MATLAB Central File Exchange), respectively. Graphs were produced using Graphpad Prism (version 7.02), and the data are shown as mean ± standard error of the mean (s.e.m.) unless stated otherwise. The results were considered statistically significant when $P \leq 0.05$. Significance levels were ns; *$P < 0.05$; **$P < 0.01$; ***$P < 0.001$; ****$P < 0.0001$ as suggested by Graphpad Prism.

**Reporting summary**. Further information on research design is available in the Nature Research Reporting Summary linked to this article.

## Data availability
The source data underlying Figs. 1a, 1b, 2a, 2c, 2f, 3a, 3b, 3c, 3d, 3e, 3f, 4b, 4c, 5, 6a, 6c, 6d, 7f–j, 7k–m, and Supplementary Figs. 1, 2, 7, 8, 10 are provided as the Source Data file. The sequence data that support the findings of this study have been deposited in the NCBI sequence read archive (SRA) database with links to BioProject accession ID PRJNA550004. All other data are available from the authors upon reasonable requests.

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

## Acknowledgements

We thank the support of members of the Whole-Genome Research Core Laboratory of Human Diseases, and the Cancer Vaccine and Immune Cell Therapy Core Laboratory, Chang Gung Memorial Hospital, Taiwan. This work was supported in part by grants from the Ministry of Science and Technology, Taiwan (MOST 102-2314-B-010-014-MY3, 103-2321-B-182-001, 104-2314-B-182A-148-MY3, 104-2325-B-182A-006, 104-2320-B-010-036-MY3, 105-2628-B-010-007-MY3, 106-2314-B-182A-037-MY3), and Chang Gung Memorial Hospital (CLRPG2E0051~3, CORPG3F0041~2, OMRPG3E0041, CMRPG1F0111~2, CORPG3F0061~2, CIRPG3I0041). Most funds for the European RegiSCAR group came from a research grant from the European Commission (QLRT-2002-01738). Maja Mockenhaupt received the Else Kröner Memorial Stipendium for support of clinical research through the Else Kröner-Fresenius-Foundation. Methodological costs were partly funded by the Deutsche Forschungsgemeinschaft (FOR 534). E.P. and S.M. received funding from National Institutes of Health (1P50GM115305-01, R21AI139021 and R34AI136815) and the National Health and Medical Research Foundation of Australia. W.T. received funding from the Faculty of Medicine, Khon Kaen University, Thailand (IN61233 and IN61301) and National Science and Technology Development Agency, Thailand.

## Author contributions

R.-Y.P. performed the experiments in the paper. M.-T.C. and C.-W.W. helped the experiments and data analysis. Y.-S.L. conducted TCR NGS data analysis and statistical analysis. F.L. generated the HLA-transgenic mice. A.W.M. produced TCR transfectants. R.S. and D.A.O. performed in silico computer modeling analysis. C.-B.C., M.M., T.B., W.T., J.-C.R and W.-H.C. enrolled patients and provided human tissue specimens and intellectual input. E.J.P., S.A.M., and K.D.W. helped TCR NGS data analysis. W.-H.C. and S.-I.H. conceived of and guided the study.

## Additional information

**Competing interests:** The authors declare no competing interests.

