## [Peer Review File · Nature Communications]

Reviewers' comments:

Reviewer #1, structural biology expert (Remarks to the Author):

In this article by Pan et al., a novel public $\alpha\beta$ TCR have been identified in CBZ-induced patients. The authors have performed SPR- analyses as well as molecular modeling to reveal the selectivity of the interaction between HLA-CBZ-TCR.

Major concerns:

1. To study the selectivity of drug TCR interactions applying SPR, the TCR needs to be very pure, to avoid unspecific binding. Thus, the authors need to show that the TCRs produced are pure by Coomassie stained SDS-PAGE (not only Western-blot, as shown in S8) and folded (size exclusion chromatography-analyses). Potentially a positive control (MHC?) could be used in the SPR analyses to further confirm activity of the immobilized proteins.
2. The SPR-curves for CBZ should be shown.
3. The authors state that the in silico model reveals that the public $\alpha\beta$ TCR directly interacts with CBZ (S10), this needs to be validated experimentally. For instance, site-directed mutagenesis of crucial residues for the CBZ binding on TCR could be substituted and re-analyzed in SPR for CBZ binding.
4. The authors refer to Figure 5B (page 12) which is lacking.

Reviewer #2, TCR repertoire expert (Remarks to the Author):

The manuscript "Selective and public T cell receptor play a critical pathogenic role in severe drug hypersensitivity" by Pan et al. analyzing the relationship between T cell receptor repertoire and drug hypersensitivity. Using T cell repertoire sequencing for the discovery of drug specific receptors in different population and validation of the found receptor for drug specificity is a promising approach. Analyzing the specificity of TCR for specific antigens an ongoing challenge and the presented results suggest that a specific receptor (TCR alpha and beta) could be identified and sufficient validated.

Some questions focusing on the TCR discovery part:

Samples:

Is there any bias in age or gender in the observed donors per group?

I couldn't find any detailed information about the 44 healthy donors? Table S1 only contains data from the patients. Especially their HLA type would be of interest and how they are comparing to the patient cohort. Along this line would it be possible to make the comparison between patients and healthy controls HLA matched?

TCR Sequencing.

The used method based on a 2010 published protocol which is quite old for repertoire sequencing. I

am wondering if e.g. UMI were introduced in the recent protocol to avoid PCR bias? PCR bias is especially prevalent if low number of cells were used as starting point for mRNA purification. I would also like to get an overview about number of cells used per sample and the amount of reads per sample (all samples) and especially unique sequences (or V-CDR3-J recombination). The manuscript mentioned always the relative frequency of CDR3 or V genes, but it is unclear on which numbers this is based on. In general, it would be helpful to provide more details to the TCR sequencing and primary analysis part.

Are the used PBMC samples separated for CD4 and CD8 cells or any other cell sub-types?

It is also not stated where the sequences can be found or if at all they will be made publicly available.

Phylogenetic tree analysis manuscript line 204 and Supplement:

Phylogenetic tree analysis for TCR is misleading and dangerous. T cell receptors can't be analyzed like genomic gene mutations and have a completely different origin (somatic VDJ recombination). There potential similarity in CDR3 is not based on nucleotide substitution on which this tree and the used model is based on. The cluster information can't be used.

Different papers are written aiming to find and determine similarities in binding affinity of TCRs (e.g. Nature 2017 published, TCRDist or GLIPH) but don't need to be applied for this study. Similarities of found CDR3 can also be origin of Sequencing error especially if one dominant clone exist which seems to be the case here.

Figure 2

A

Can you provide an error bar to estimate the differences between the donors?

B and C

Did both figures show the same data? B from one person and C from all 7 combined in a different plotting method?

The differences in observed frequency of the "public" TCR differs a lot between the patients especially in the PBMC samples from the CBZ-SJS/TEN PBMC. Is the range really from 0.0 to 4.2%? Can you explain more the observed differences or suggested causes?

The 4.2% of all PBMC T cells seems to be very high. Is it correct that the PBMC samples contain all T cells types? CD4, CD8, naïve and memory etc. and still up to 4.2% of the reads (cells ?) are from one specific CDR3? The manuscript stated that in blister cells it is mainly memory CD8pos cells with the TCRs of interest. Is this suggesting that very high percentage of memory CD8 cells contain just one CDR3 to have an overall frequency of 4.2% of all T cells. Please clarify.

Not knowing the exact procedure of the TCR sequencing is it possible that the donors were processed and sequenced together using the different barcodes which may let to false positive findings in other patient samples due to errors in the barcode sample assignments? My questions are aiming to address the question to which degree it is possible to generalize the finding of the "public" TCR for diagnostic is given the low number of patients and the huge variation of their TCR frequency particularly in their PBMCs (Figure 3ab).

Outlook discussion. Would it be possible to predict who has a higher chance to develop the hypersensitivity?

Overall the provided data support the claim that a TCR clonotype for a specific drug hypersensitivity were identified. The presented results are novel and of interest for other researchers.

Reviewer #3, clinical drug hypersensitivity expert (Remarks to the Author):

Manuscript background information

Drug hypersensitivity reactions (DHR) are drug induced inflammatory reactions that extend from mild reactions like urticaria mediated by IgE or maculo-papulous erythema mediated by T cells to life threatening diseases like Stevens Johnson Syndrome (SJS), Toxic epidermo-necrolysis (TEN) or DRESS (Drug Rash with eosinophils and systemic symptoms). Fortunately these severe diseases are rare. Nevertheless, since the pathomechanism is not well understood, the only appropriate treatment is the rapid arrest of the drug, jeopardizing the chances for the affected patients to recover or even to survive. For such reasons, there is an urgent need to better understand the pathomechanisms of DHR, especially for the severe reactions. In the last 10 years, progress has been done with the identification of risk alleles, particularly in the HLA loci that increase the susceptibility to develop severe DHR. Whereas such associations can reach notable values like 100% in a given ethnicity, the positive predictive values can be very low (a few percent), suggesting that other factors are involved. Such a factor could be a define TCR repertoire. In the submitted study, as other factor candidate, TCR sequences were analyzed by next generation sequencing at the level of skin lesions as well as in the peripheral blood. This analysis was done in an impressive cohort including 65 patients with SJS/TEN caused by carbamazepine. Other patients with SJS/TEN caused by other drugs were included, as well as drug tolerant and healthy controls. The HLA-B locus of the cohort individuals has been typed and published in the suppl. table 1, however no indication is provided about the other HLA class I loci. Given that 13 patients have Caucasian ethnicity, the typing of the HLA-A and C loci would be appreciated, since carbamazepine induced SJS/TEN has been shown to be associated with HLA-A*31:01 in Caucasians.

The TCR sequencing revealed striking overexpression of TRBV12-4 paired with TRBJ2-2 in patients with cbz induced SJS/TEN. A particular clonotype shared among cbz induced SJS/TEN patients of different ethnicities could be identified in the β -chain, even in HLA-B*15:02 negative patients. This clonotype was called therefore "public TCR" by the authors. Of note this public TCR is different from another clonotype identified by the same authors in another study on cbz induced TEN a few years ago. Concerning the alpha chain of the TCR, one dominant clonotype could be identified in the case 4. Unfortunately whether this dominant alpha chain clonotype is also over-expressed in other patients has not been addressed. T cell cloning in all the patients would be a tedious to answer this question, but sequencing of the alpha chain or TaqMan real time PCR (as for figure 3e, in the case of the β -chain) could be quickly helpful.

Then the involved TCR was produced in soluble form and spr was applied to study the binding capacities of drugs onto this TCR. The provided information about the construction of the single chain construct in the methods part is quite vague. The orientation of the α and β chain, the sequence of the spacer, the inclusion of the constant part of the TCR and the arrangement of the Fc region and its size or simply the entire sequence should be published. Only carbamazepine and its derivation showed some affinity to the construct. Other drugs were not able to bind. The authors performed some docking calculations as well, supposed to be shown of fig. 5B, although there was no fig 5 b in the given manuscripts (maybe fig. S10?). Moreover, no word (!) about the docking experiments could be found in the methods part. Then TCR transfectants were generated with a murine 5KC hybridoma background. Again the sequence and the hybridization (human-murine constant part) should be published. In figure 6, TCR transfectants were shown to be activated by carbamazepine and derivative. Although, the responses are statistically significant and the specificity of the reactivity is shown without any doubts, the reactivity seems to be quite low. According to the methods section, 105 cells were plated per well and only up to 300 cells could be activated and produced IL2, corresponding to 0.3 % of a monoclonal population. This aspect and possible explanations could be addressed in the discussion section.

Finally, the authors were able to develop a remarkable animal model for cbz induced TEN. This model functions only in presence of HLA-B*15:02 tg mice and T cells expressing the identified public TCR. These findings represent an important breakthrough in the field of DHR, which is terribly lacking useful animal models. Interestingly, the affected animal exhibited lesions not only in the skin but also in

internal organs, which is more typical for DRESS.

In the submitted study, the authors focused mainly on cbz induced SJS/TEN (and oxc), for which they have the highest numbers on patients to study. For the other drugs, the identification of a public TCR shared by most patients was more complicated. For allopurinol for example, dominant clonotypes were found in 2 patients, but the clonotypes "ASSQDLTGNTI" and "ASSPRDFSYEQY" cannot be considered homologous. In the drug allergy field, people have the tendency to generalize their finding to all other hypersensitivity reactions induced by other drug. For years, haptenization has been thought to be an essential driver of DHR. When the presentation of altered peptide repertoire on HLA-B*57:01 in the presence of abacavir was discovered, every DHR was thought to be mediated by altered peptide repertoire presentation. In this study, the findings are important and represent a breakthrough for cbz induced SJS/TEN. But it is not sure yet, whether the public TCR could be generalized to other SJS/TEN caused by other drugs. This position should be discussed in the discussion.

Altogether, this study represents a real breakthrough because of two important aspects. First a public TCR could be identified in patients of different ethnicities (HLA) with cbz induced SJS/TEN. It was thought for very long time, that particular TCR sequences in the TCR repertoire might represent a risk for the development of severe DHR. Here the authors could prove this assumption for the first time. Secondly, an animal model for SJS/TEN could be generated that is at last not based on the hapten theory.

Other points:

- Suppl table 1

Cbz induced TEN in Caucasian have been associated with the allele HLA-A*31:01. If the alleles of the HLA-A and C are also known, it would be important to communicate them.

- Figure 2

The figure is not clear enough. The letters are barely readable.

- Figure S7

Please notify the color of the α and β chains. Is the pink segment coming from the α or from the β chain?

- Figure 5

Section B is missing, as well as the docking part in the method section

- Soluble TCR and TCR transfectants

Please provide the complete sequence of all the constructs.

- Lane 67-68. The sentence is not grammatically correct

Nature Communication

Manuscript Number: NCOMMS-18-25789

Manuscript Title: Characterization of drug-specific public TCR and adoptive T cell transfer to HLA transgenic mice confer the immune synapse of severe cutaneous adverse reactions

Reviewers' comments:

Reviewer #1, structural biology expert (Remarks to the Author):

In this article by Pan et al., a novel public $\alpha\beta$ TCR have been identified in CBZ-induced patients. The authors have performed SPR- analyses as well as molecular modeling to reveal the selectivity of the interaction between HLA-CBZ-TCR.

Major concerns:

1-1. To study the selectivity of drug TCR interactions applying SPR, the TCR needs to be very pure, to avoid unspecific binding. Thus, the authors need to show that the TCRs produced are pure by Coomassie stained SDS-PAGE (not only Western-blot, as shown in S8) and folded (size exclusion chromatography-analyses). Potentially a positive control (MHC?) could be used in the SPR analyses to further confirm activity of the immobilized proteins.

Reply: We've generated the scTCR-Fc recombinant protein from HEK293F cells transfected with scTCR-Fc vector and purified the protein by protein A beads as described in the **Materials and Methods (Page 18)**

“.... The scTCR-Fc plasmid was transfected into the HEK293F cells, and the soluble single chain scTCR-Fc recombinant protein was purified from the culture medium by protein A beads⁵⁸.”

The purity of the purified recombinant protein was examined by the SDS-PAGE and SYPRO® Ruby staining shown in the **Figure S9a (Supplement Materials, page 15)**. The molecular weight of scTCR-Fc recombinant protein is approximately 70kDa. The SDS-PAGE and SYPRO Ruby staining show that the scTCR-Fc recombinant protein with a molecular weight of approximately 70kDa accounted for the major protein in the serum-free cultured medium of the culture supernatant of HEK293F cells transfected with scTCR-Fc plasmid (**Figure S9a**). The scTCR-Fc recombinant protein was purified by protein A beads, and analyzed by the SDS-PAGE and SYPRO Ruby staining. The proportion of scTCR-Fc in the culture medium (**Figure S9a**, lane 2) and purified recombinant protein (**Figure S9a**, lane 6) was estimated by image J software, which showed 92.4% and 98%, respectively (**Figure S9a**).

Figure S9a

Since the proportions of scTCR-Fc in the culture medium or the purified recombinant protein eluted from protein A beads were very high, we analyzed the binding response and activity of the protein samples from both sources by SPR analysis. As shown in **Figure S10**, the binding response of the protein from either source was similar. Our data suggest that the scTCR-Fc protein possesses binding activity towards carbamazepine and its structural analogs.

1-2. The SPR-curves for CBZ should be shown.

Reply: Thank you. The SPR-curves for CBZ are shown in the **Figure S10 (Supplement Materials, page 16)**. As the proportions of scTCR-Fc in the culture medium or the purified recombinant protein eluted by protein A beads were very high, we analyzed the binding response and activity of the protein samples from both sources by Biacore T200 SPR assay. The **Figure S10a** shows the binding response observed using the protein sample directly from the culture supernatant of the public scTCR-Fc plasmid-transfected HEK293F cells. The **Figure S10b** exhibits the binding response of the scTCR protein purified by the protein A beads. The drug flowed through the chip, and the binding response was measured. The binding response of the protein from either source was similar and very low. The measured K_d values of scTCR-Fc protein from culture supernatant and eluted from protein A beads were 5.49×10^{-3} M and 1.6×10^{-2} M, respectively.

1-3. The authors state that the in silico model reveals that the public $\alpha\beta$ TCR directly interacts with CBZ (S10), this needs to be validated experimentally. For instance, site-directed mutagenesis of crucial residues for the CBZ binding on TCR could be substituted and re-analyzed in SPR for CBZ binding.

Reply: Thanks for the comment. We agree that the experiments of site-directed mutagenesis can further validate the key residues of TCR for CBZ binding. However, our SPR data showed that the binding affinity of TCR to the CBZ drug antigen was very low (approximately $10^{-2} \sim 10^{-3} \text{ M}$) (Figure S10), and amino-acid substitutions were observed in the “CDR3 ASSLAGELF cluster” of the public TCR β in the blister samples of CBZ-SJS/TEN (Figure 2f, Figure S4). We tried to generate TCR recombinant proteins with single amino-acid substitution; however, we did not have a sensitive assay system to identify the crucial residues of CDR3 ASSLAGELF of the public TCR for CBZ binding. The low affinity of TCR may be related to the fact that a chemical drug is small and much difficult to be a good antigen compared to the peptide antigen.

In Figure 5, we showed that the public TCR possessed binding affinity to CBZ and its structural analogs (CBZ-epoxide, OXC, etc.), but not other drugs. In addition, the Figure S11 shows that the control $\alpha\beta$ TCR recombinant protein had no binding response to CBZ or other

drugs. Although the public TCR showed binding response to CBZ, the affinity and immune response of TCR itself were weak (**Figure 5**, D condition of **Figure 6a**). Our TCR transfectants study revealed that the specific TCR clonotype triggered significant T cell activation in the presence of CBZ and HLA-B*15:02 (F condition of **Figure 6a**). Our HLA-B*15:02 transgenic mice further showed the animals developed phenotypes mimicking to the clinical presentations of CTL-mediated drug hypersensitivity after received the adoptive cell transfer of the specific TCR-transfected T lymphocytes with oral administration of CBZ (**Figure 7**). Based on these results, we confirm that the public $\alpha\beta$ TCR indeed interacts with CBZ, and the immune response is promoted and becomes significance by the formation of immune synapse composed of HLA-drug antigen-TCR.

1-4. The authors refer to Figure 5B (page 12) which is lacking.

Reply: Thanks for your reminder. It was a typo error for Figure 5B, and we already corrected and changed it to Figure S12 in the revised manuscript.

The results of docking experiments are shown in **Figure S12**.

Figure S12. *In silico* molecular modeling of the interaction of the public $\alpha\beta$ TCR, carbamazepine, and HLA-B*15:02 protein. The crystal structure of HLA-B*15:01 (PDB code 1XR8) was adopted for the model of HLA-B*15:02, and the molecular docking software (AutoDock Vina) was applied to predict CBZ binding to the HLA-B*15:02/peptide/TCR complex. The highest scoring interaction was identified between CBZ and a site comprised of the α 1 helix of HLA-B*15:02, TCR α CDR3 VFDNTDKLI, and TCR β CDR2 (ΔG -7.9 kcal/mol).

Reviewer #2, TCR repertoire expert (Remarks to the Author):

The manuscript “Selective and public T cell receptor play a critical pathogenic role in severe drug hypersensitivity” by Pan et al. analyzing the relationship between T cell receptor repertoire and drug hypersensitivity. Using T cell repertoire sequencing for the discovery of drug specific receptors in different population and validation of the found receptor for drug specificity is a promising approach. Analyzing the specificity of TCR for specific antigens an ongoing challenge and the presented results suggest that a specific receptor (TCR alpha and beta) could be identified and sufficient validated.

Some questions focusing on the TCR discovery part:

Samples:

2-1. Is there any bias in age or gender in the observed donors per group?

Reply: Thanks for your comment. Because SCAR is a rare disease, there were difficulties to collect proper and well-preserved biological samples (e.g., blister cells, PBMC of the active or recovery stage) with a large sample size for the TCR RNA sequencing and single cell assay. In addition, HLA genetic predisposition is well-known in the disease of SCAR caused by some culprit drugs, e.g., carbamazepine, oxcarbazepine, and allopurinol. This study was designed to prospectively enroll the patients, and recruit controls based on the information of drug history, drug tolerance, and preferred HLA alleles. Enrolling age/gender-matched controls was not the first criteria of this study.

In this study, our case group is composed of 73 patients of SCAR, including 8 cases with CBZ-DRESS, and 65 with SJS/TEN caused by carbamazepine (CBZ) (n = 42), oxcarbazepine (OXC) (n = 3), lamotrigine (LTG) (n = 4), phenytoin (PHT) (n = 6), and allopurinol (n = 10) (**Table S1**). The average age of the CBZ-SJS/TEN patients was 46.86 (range from 23-84 yrs), and the sex distribution ratio was F:M=16:13 (**Table S1**). As patients with CBZ-SJS/TEN accounted for the major population of our cases, and the public TCR was identified from this group, we enrolled drug-tolerant controls (n=12) who had taken CBZ for more than 6 months without adverse reactions, and 6 of them carried *HLA-B*15:02* (**Table S2**). The average age of the CBZ-tolerant controls was 45.75 (range from 20-83 yrs), and the sex distribution ratio was F:M=6:6 (**Table S2**). The age and gender of case/control were matched in our study. In addition, we recruited 44 healthy donors to represent the general population, which showed the phenotype frequency of *HLA-B*15:02* as 9% (**Table S3**).

2-2. I couldn't find any detailed information about the 44 healthy donors? Table S1 only contains data from the patients. Especially their HLA type would be of interest and how they are comparing to the patient cohort. Along this line would it be possible to make the comparison between patients and healthy controls HLA matched?

Reply: Thanks for the comment. We have provided the HLA genotype results of SCAR patients, tolerant controls, and healthy donors in Table S1, S2, and S3 in the revised manuscript.

As *HLA-B*15:02* is known to be strongly associated with CBZ-SJS/TEN and the public TCR was identified from patients of this group, we examined the presence of *HLA-B*15:02* allele in the subjects of CBZ-tolerant controls and selected them for this TCR repertoire study. Among the 42 cases of CBZ-induced SJS/TEN, *HLA-B*15:02* allele was found in all 24 (100.00%) of Chinese, 4 of 5 (80.00%) Thai patients, and 2 of 13 (15.38%) subjects enrolled from Europe (**Table S1**). In addition, all three patients with OXC-SJS carried *HLA-B*15:02* (**Table S1**). Among the 12 subjects of CBZ-tolerant controls, 6 (50%) carried *HLA-B*15:02* (**Table S2**). We also recruited 44 healthy donors to represent the general population, which had the phenotype frequency of *HLA-B*15:02* as 9% compatible to the published information (**Table S3**). The information has been added to the **Results** (Page 7).

The public TCR CDR3 clonotype identified in this study was abundant in the PBMC (0.1-3.6%) of CBZ-SJS/TEN patient of Han Chinese, Thai people, and Europeans those most had no *HLA-B*15:02* allele (**Figure 3a-e**). Furthermore, the expression of this specific clonotype was predominately in CBZ-SJS/TEN, but absent in CBZ-tolerant controls or healthy donor controls who carried *HLA-B*15:02* allele (**Figure 3c-3e**). Our data suggest that the presence of the public TCR is correlated with the disease phenotype CBZ-SJS/TEN, and a *HLA-B*15:02*-favored but unrestricted manner.

2-3. TCR Sequencing.

The used method based on a 2010 published protocol which is quite old for repertoire sequencing. I am wondering if e.g. UMI were introduced in the recent protocol to avoid PCR bias? PCR bias is especially prevalent if low number of cells were used as starting point for mRNA purification.

Reply: Thanks for the comment. The method of TCR repertoire sequencing in our study is based on iRepertoire® library preparation system (illumina) and single cell sequencing based on the paper published in Nature Biotechnology in 2014 [ref.33]. Rosati *et al.* (BMC Biotechnology, 2017, 17(1):61) have compared the different library preparation system, including iRepertoire®-based PCR, 5' RACE-based PCR and UMI-corrected 5' RACE based PCR, and showed that the percentages of sequences captured by these three methods are similar when considering the highly abundant clonotypes (i.e., clonotypes are almost the same) (see the following Table from that paper). We added these to the **Discussion** section (Page 15).

Table 2 Percentages of CDR3 nucleotide sequences detected in both duplicates of the same method

Replicates shared clonotypes percentages	α chain			β chain		
	iRepertoire	5'RACE	5'RACE + UMI	iRepertoire	5'RACE	5'RACE + UMI
All clonotypes	36	44	20	35	52	25
Top 300 clonotypes	31	26	27	35	32	37
Top 100 clonotypes	37	31	36	46	51	51
Top 50 clonotypes	45	44	50	38	64	64
Top 20 clonotypes	60	65	70	50	80	75

The percentages are shown for comparisons made between all of the observed clonotypes and between the 300, 100, 50 and 20 most abundant sequences detected by each method. Results include iRepertoire kit data, 5'RACE-based PCR data and data from the same PCR corrected using unique molecular identifiers. Data are shown for both α and β chains

(From Rosati et al. BMC Biotechnology (2017) 17(1):61)

We tried to apply the UMI system of 10X Genomics for single cell TCR sequencing on the blister samples of patients with SJS/TEN. However, we failed due to the limited source of fresh samples, and high amounts of necrotic keratinocytes in the samples also interfered and failed from the initial quality control of 10X Genomics system. Thanks for the comments. We are conducting experiments to introduce UMI to our following studies.

In addition to the NGS data, the expression of our public TCR was also confirmed by qPCR assay and single-cell sequencing (see **Figure 3e** and **Figure 4**). We further performed functional assays (e.g. SPR, TCR transfectants assay, and *HLA-B*15:02* transgenic mouse model) to confirm the role of the TCR clonotype in SCAR. According to the data of the NGS sequencing on the large sample size of blister cells and PBMC samples of patients enrolled from different populations and *in vitro/in vivo* functional studies, our results confer the essential role of the public TCR clonotype in SCAR.

2-4. I would also like to get an overview about number of cells used per sample and the amount of reads per sample (all samples) and especially unique sequences (or V-CDR3-J recombination). The manuscript mentioned always the relative frequency of CDR3 or V genes, but it is unclear on which numbers this is based on. In general, it would be helpful to provide more details to the TCR sequencing and primary analysis part.

Reply: The absolute numbers of total reads per sample (all samples) and the preferential TCRβ CDR3 sequences and reads of the blister cells of patients with SJS/TEN are listed in the **Table 1**. The TCR sequencing reads per sample are around 100,000~1,300,000 (**Table 1**). We used 100-500ng of total RNA of each sample of blister cells or PBMC to convert to cDNA and produce the library by multiple PCR amplifications using a panel of TCR primers specifically targeted to the V, D, and J gene regions, and sequenced using an Illumina MiSeq system (**Materials and Methods, Page 18**). As one typical mammalian cell contains 10–30 pg total RNA (Reference:

<https://www.qiagen.com/ie/resources/faq?id=06a192c2-e72d-42e8-9b40-3171e1eb4cb8&lang>

en), we estimate that the amounts of 100-500 ng of total RNA per sample we used in our study represented approximately 3,333~50,000 cells per sample.

We also performed single cell sequencing using the protocol published in Nature Biotechnology 2014 [ref.33]. The absolute reads per sample of 30 single cells are listed below. Single cells from blister samples were sorted by flow cytometry, and the reads and sequences of TCR α /TCR β and 18 functional genes with granulysin (GNLY) in the panel were determined (see **Materials and Methods, Page 18**). All of the 30 cells expressed the same specific TCR β CDR3 clonotype “ASSLAGELF”, and twenty-five cells expressed the TCR α CDR3 clonotype “VFDNTDKLI”. The TCR α and TCR β sequencing reads per sample are around 10~10,000 (**Table 1**).

	TCR β	TCR α		
	CDR3: ASSLAGELF	CDR3: VFDNTDKLI	CDR3: AASPPDGNQFY	CDR3: ALDIPNFGNEKLT
cell 1	18	0	26	0
cell 2	13	0	20	0
cell 3	28	8	0	0
cell 4	211	16	0	0
cell 5	868	4867	0	0
cell 6	329	66	0	0
cell 7	10	13	0	0
cell 8	230	93	0	0
cell 9	222	97	0	0
cell 10	13	25	0	0
cell 11	11	18	0	0
cell 12	13	71	0	0
cell 13	37	0	16	0
cell 14	168	20	0	0
cell 15	31	0	0	17
cell 16	27	36	0	0
cell 17	43	27	0	0
cell 18	379	75	0	0
cell 19	1225	52	0	0
cell 20	161	32	0	0
cell 21	47	525	0	0
cell 22	69	132	0	0
cell 23	17	15	0	0
cell 24	26	14	0	0
cell 25	16	0	22	0
cell 26	10785	22	0	0
cell 27	7719	355	0	0
cell 28	67	124	0	0
cell 29	51	3275	0	0
cell30	47	24	0	0

2-5. Are the used PBMC samples separated for CD4 and CD8 cells or any other cell sub-types?

Reply: We didn’t separate the PBMC samples for any specific T cell sub-types. We had separated and analyzed specific T cell sub-types from the blister cells of SJS/TEN skin lesions, and found that the public TCR clonotype was predominately expressed by the memory CTL (CD8⁺CD45RA⁻CD197⁻) (**Figure 2g, Figure S5**).

2-6. Figure 2g

It is also not stated where the sequences can be found or if at all they will be made publicly available.

Reply: The sequences of the paired TCRa/b identified in this study are shown in **Figure S8**.

2-7. Phylogenetic tree analysis manuscript line 204 and Supplement:

Phylogenetic tree analysis for TCR is misleading and dangerous. T cell receptors can't be analyzed like genomic gene mutations and have a completely different origin (somatic VDJ recombination). Their potential similarity in CDR3 is not based on nucleotide substitution on which this tree and the used model is based on. The cluster information can't be used.

Different papers are written aiming to find and determine similarities in binding affinity of TCRs (e.g. Nature 2017 published, TCRDist or GLIPH) but don't need to be applied for this study. Similarities of found CDR3 can also be origin of Sequencing error especially if one dominant clone exists which seems to be the case here.

Reply: Thanks for the comment. The phylogenetic tree analysis was deleted in the revised manuscript.

2-8. Figure 2a

Can you provide an error bar to estimate the differences between the donors?

Reply: We've provided the error bars and modified the **Figure 2a**, accordingly.

(Figure 2a):

2-9. Figure 2b and 2c

Did both figures show the same data? 2b from one person and 2c from all 7 combined in a different plotting method?

Reply: Thank you. Yes, the figure 2b, 2c, and 2d were from the same dataset of the blister cells of 7 patients with CBZ-SJS/TEN. The figure 2b shows the representative data of TRBV/TRBJ pairings of one CBZ-SJS/TEN patient (case 4). The figure 2c and 2d are the data from all 7 CBZ-SJS/TEN patients (case 1~case 7). Figure 2c presents the mean frequencies

of each *TRBV/TRBJ* pairings of blister cells from CBZ-SJS/TEN patients (n=7). Figure 2d is the pairwise overlap Circos plot, which shows the overlapping CDR3 clonotypes among these 7 samples. We've revised the figure legends, accordingly.

2-10. The differences in observed frequency of the “public” TCR differs a lot between the patients especially in the PBMC samples from the CBZ-SJS/TEN PBMC. Is the range really from 0.0 to 4.2%? Can you explain more the observed differences or suggested causes?

The 4.2% of all PBMC T cells seems to be very high. Is it correct that the PBMC samples contain all T cells types? CD4, CD8, naïve and memory etc. and still up to 4.2% of the reads (cells ?) are from one specific CDR3? The manuscript stated that in blister cells it is mainly memory CD8pos cells with the TCRs of interest. Is this suggesting that very high percentage of memory CD8 cells contain just one CDR3 to have an overall frequency of 4.2% of all T cells. Please clarify.

Reply: We didn't separate the PBMC samples for any specific T cell sub-types. Our PBMC samples contained all T cell types. We did observe that the frequency of the “public” TCR differs between the samples and patients. Because our samples were collected from different labs/hospitals of different countries, there were uncontrolled issues. The differences of the frequencies of the public TCR in our samples may relate to (1) the various states of sample collection during the disease (e.g., early phase, acute stage or recovery stage), (2) the disease severity (i.e., SJS has skin detachment less than 10% of total body surface area (TBSA), SJS-TEN overlap has skin detachment of 10-29% TBSA, and TEN has skin detachment more than 30% TBSA), (3) the methods of sample preservation, and (4) the methods of treatment (e.g., supporting care, steroids, cyclosporine, anti-TNF-alpha biologics, or IVIG). We checked the clinical data of patients (case 4 and case 5), who had high frequencies of the public TCR, and found that these were diagnosed as SJS-TEN overlap or TEN, and the samples were collected at acute stage.

2-11. Not knowing the exact procedure of the TCR sequencing is it possible that the donors were processed and sequenced together using the different barcodes which may let to false positive findings in other patient samples due to errors in the barcode sample assignments? My questions are aiming to address the question to which degree it is possible to generalize the finding of the “public” TCR for diagnostic is given the low number of patients and the huge variation of their TCR frequency particularly in their PBMCs (Figure 3ab).

Reply: We used the barcode system for sequencing our samples of cases, tolerant controls, and the healthy donors. The case and control samples were usually processed and sequenced together using the different barcodes which were located in the both ends of the PCR products (reads) of each sample. We have established a series of quality control procedures to

exclude the false assignments of samples, and eliminate the low-quality sequencing reads. The QC-passed sequence reads have matched specific forward and reverse barcode sequences. The detailed TCR analysis method was described in the **Materials and Methods**. We didn't find the public TCR with >0.001% in the samples of HD or tolerant controls. Furthermore, our qPCR assays confirmed the NGS data.

As the reviewer pointed out, the patient number remains low and frequency of this public TCR differs among samples. Although the public TCR identified in this study has potential to be used for the diagnosis, there are issues needed to be overcome. The observation of diverse frequencies in the samples of CBZ-SJS/TEN patients may relate to the uncontrolled factors discussed in our replies to question 2-10. In addition, as TCRs were generated by somatic rearrangement, the course of disease affects the cell populations/numbers, and the amounts of transcripts expressing the TCR clonotypes. Furthermore, the technologies applied for detecting the public TCR for diagnosis purpose have different degree of sensitivity. For example, NGS has high sensitivity to detect the low number of TCR reads in the PBMC samples, but the rapid qPCR assay has difficulties to precisely measure the specific TCR clonotypes in the samples collected from patients in the recovery stage.

2-12. Outlook discussion. Would it be possible to predict who has a higher chance to develop the hypersensitivity?

Reply: The germline variants of *HLA* and *CYP* alleles have been linked to SCAR, and some of the genetic markers have been applied in clinics to prevent the drug hypersensitivity reactions (See the **Introduction page 7** and **Discussion page 14**). However, both the *HLA* and *CYP* genetic variants have low positive predictive values (e.g., the PPV of *HLA-B*15:02* is ~3% for CBZ-SJS/TEN). In the general population of Han Chinese, there are 9% of people carrying *HLA-B*15:02*, yet the incidence of CBZ-SJS/TEN is only 0.3%, suggesting other factors are required for the development of SCAR reactions.

In this study, we discover that the public TCR is another key factor for the development of SCAR. We found the public TCR showed drug- and phenotype-specific patterns, and a *HLA-B*15:02*-favored but unrestricted manner in the patients enrolled from different ethnic populations. Addition of TCR to the panel of genetic test shall increase the positive predictive value (PPV) and the sensitivity to the disease diagnosis and prevention. Further studies are required to evaluate how these results could be applied to clinics to diagnose, predict and prevent the fatal hypersensitivity reactions and to develop new therapeutics for the disease.

2-13. Overall the provided data support the claim that a TCR clonotype for a specific drug hypersensitivity were identified. The presented results are novel and of interest for other researchers.

Reply: Thanks.

Reviewer #3, clinical drug hypersensitivity expert (Remarks to the Author):

Manuscript background information

Drug hypersensitivity reactions (DHR) are drug induced inflammatory reactions that extend from mild reactions like urticaria mediated by IgE or maculo-papulous erythema mediated by T cells to life threatening diseases like Stevens Johnson Syndrome (SJS), Toxic epidermo-necrolysis (TEN) or DRESS (Drug Rash with eosinophils and systemic symptoms). Fortunately these severe diseases are rare. Nevertheless, since the pathomechanism is not well understood, the only appropriate treatment is the rapid arrest of the drug, jeopardizing the chances for the affected patients to recover or even to survive. For such reasons, there is an urgent need to better understand the pathomechanisms of DHR, especially for the severe reactions. In the last 10 years, progress has been done with the identification of risk alleles, particularly in the HLA loci that increase the susceptibility to develop severe DHR. Whereas such associations can reach notable values like 100% in a given ethnicity, the positive predictive values can be very low (a few percent), suggesting that other factors are involved. Such a factor could be a define TCR repertoire. In the submitted study, as other factor candidate, TCR sequences were analyzed by next generation sequencing at the level of skin lesions as well as in the peripheral blood. This analysis was done in an impressive cohort including 65 patients with SJS/TEN caused by carbamazepine. Other patients with SJS/TEN caused by other drugs were included, as well as drug tolerant and healthy controls.

3-1. The HLA-B locus of the cohort individuals has been typed and published in the suppl. table 1, however no indication is provided about the other HLA class I loci. Given that 13 patients have Caucasian ethnicity, the typing of the HLA-A and C loci would be appreciated, since carbamazepine induced SJS/TEN has been shown to be associated with HLA-A*31:01 in Caucasians.

Reply: Thanks. We've the information of HLA-A data in the **Table S1**.

The HLA-A, HLA-B, and HLA-C genotypes of the 13 cases of the European patients with CBZ-SJS/TEN are shown below.

HLA-A, HLA-B, and HLA-C genotypes of the 13 cases of the European patients with CBZ-SJS/TEN

Patient ID	Culprit drug	Clinical diagnosis	HLA-A genotype	HLA-B genotype	HLA-C genotype	Populations
Case25	CBZ	SJS	A*01:01 / A*66:01	B*07:02 / B*41:02	C*07:01 / C*17:01	Europeans
Case26	CBZ	SJS	A*02:01 / A*11:01	B*15:02 / B*44:02	C*05:01 / C*08:01	Asian descent
Case27	CBZ	SJS	A*02:01 / A*23:01	B*41:01 / B*49:01	C*07:01 / C*07:01	Europeans
Case28	CBZ	SJS	A*02:01 / A*11:01	B*15:02 / B*44:02	C*05:01 / C*08:01	Asian descent
Case29	CBZ	SJS	A*02:07 / A*34:01	B*15:21 / B*46:01	n.a.	Asian descent
Case30	CBZ	SJS	A*01:01 / A*03:01	B*08:01 / B*35:03	C*04:01 / C*07:01	Europeans
Case31	CBZ	SJS	A*01:01 / A*01:01	B*08:01 / B*57:01	C*06:02 / C*07:01	Europeans
Case32	CBZ	SJS	A*24:02 / A*26:01	B*35:01 / B*38:01	n.a.	Europeans
Case33	CBZ	SJS	A*02:01 / A*03:01	B*07:02 / B*57:01	C*06:02 / C*07:02	Europeans
Case34	CBZ	SJS	A*01:01 / A*31:01	B*57:01 / B*57:01	C*06:02 / C*07:01	Europeans
Case35	CBZ	SJS	A*01:01 / A*30:02	B*08:01 / B*55:01	C*04:01 / C*07:01	Europeans
Case36	CBZ	SJS	A*01:01 / A*31:01	B*13:01 / B*57:01	C*05:01 / C*06:02	Europeans
Case37	CBZ	SJS	A*01:01 / A*66:01	B*08:01 / B*41:02	n.a	Europeans

3-2. The TCR sequencing revealed striking overexpression of TRBV12-4 paired with TRBJ2-2 in patients with cbz induced SJS/TEN. A particular clonotype shared among cbz induced SJS/TEN patients of different ethnicities could be identified in the β -chain, even in HLA-B*15:02 negative patients. This clonotype was called therefore “public TCR” by the authors. Of note this public TCR is different from another clonotype identified by the same authors in another study on cbz induced TEN a few years ago.

Reply: Thanks for the comments. We’ve added a paragraph in the **Discussion (Page 15)** for the difference of TCR data of these two studies.

“...The TCR clonotype identified in this study is different from our previous report, which utilized the traditional cloning and Sanger sequencing method with the samples of co-cultures of EBV-transformed B cells as the APC and the PBMC of CBZ-SJS/TEN patients²⁸. The *in vitro* expansion of T cells by co-culturing with EBV-transformed B cells as APC may distort the T cell repertoire²⁹. To reduce the methodological biases, herein we use the blister cells of SJS/TEN patients enrolled from a large cohort, and apply multiplex PCR of TCR subtype specific primers (i.e., iRepertoire® library preparation system) for NGS and single-cell sequencing, and validate the TCR function by transfection and adoptive T cell transfer into the HLA-B*15:02-transgenic mice. The iRepertoire®-based PCR library preparation system and single-cell sequencing method used in this study have been applied by many studies described in the literature³⁰⁻³³”

3-3. Concerning the alpha chain of the TCR, one dominant clonotype could be identified in the case 4. Unfortunately whether this dominant alpha chain clonotype is also over-expressed in other patients has not been addressed. T cell cloning in all the patients would be a tedious to answer this question, but sequencing of the alpha chain or TaqMan real time PCR (as for

figure 3e, in the case of the β -chain) could be quickly helpful.

Reply: Thanks for the comment. We designed a TaqMan real-time PCR assay for validating the data of NGS and single cell sequencing using the samples of other patients (see the **Materials and Methods, page 20**).

“...The Taqman real-time PCR assay for detecting the TCR α clonotype “VFDNTDKLI” was designed as the forward primer: 5’-CTCAGTGATTCAGCCACCTACCT-3’, reverse primer: 5’-TGGTCCCAGTCCCAAAGATG-3’, and probe: 5’-TCGATAACACCGACAAGC-3’ (Life Technology). The expression level of the specific TCR clonotypes was normalized by that of CD3, and the detection limit of the TCR clonotype/CD3 ratio was 0.0001. The number of cycles necessary to reach threshold fluorescence for each gene or β -actin control reaction was calculated at the crossing point (cycle threshold), and the cycle threshold of CD3 or β -actin in each reaction was used as the internal control in parallel experiments...”

The results of TaqMan real-time PCR assays for TCR alpha were shown in the **Results** (page 11) and **Figure S7**.

“...The expression of the specific TCR α CDR3 clonotype “VFDNTDKLI” was further validated by TaqMan quantitative real-time PCR in the blister cells (n=5) or PBMC samples (n=6) of CBZ-SJS/TEN patients, CBZ-DRESS patients (n=5), and controls (n=4) (**Figure S7**). The expression levels of TCR α CDR3 “VFDNTDKLI” and TCR β CDR3 "ASSLAGELF" show a similar trend, supporting the correct pairing (**Figure S7**)...”

Figure S7. Expression of the specific TCR α CDR3 clonotype “VFDNTDKLI” in the blister cells or PBMC samples of CBZ-SJS/TEN patients, CBZ-DRESS patients, and healthy donor (HD) controls. (a) The expression of TCR α CDR3 clonotype “VFDNTDKLI” was determined by TaqMan quantitative real-time PCR using the samples of the blister cells of CBZ-SJS/TEN patients (n=5), or PBMC of CBZ-SJS/TEN patients (n=6), CBZ-DRESS patients (n=5) and health donors (HD) (n=4). (b) The expression levels of TCR α CDR3 “VFDNTDKLI” and TCR β CDR3 "ASSLAGELF" were compared in different samples of blister cells from CBZ-SJS/TEN patients and PBMC of healthy donors (HD). Similar expression trend of CDR3 “VFDNTDKLI” and TCR β CDR3 "ASSLAGELF" was found in different blister cell samples.

3-4. Then the involved TCR was produced in soluble form and spr was applied to study the binding capacities of drugs onto this TCR. The provided information about the construction of the single chain construct in the methods part is quite vague. The orientation of the α and β chain, the sequence of the spacer, the inclusion of the constant part of the TCR and the arrangement of the Fc region and its size or simply the entire sequence should be published.

Reply: The complete sequence of the soluble single-chain TCR (scTCR-Fc) is shown in **Figure S8**.

3-5. Only carbamazepine and its derivation showed some affinity to the construct. Other drugs were not able to bind. The authors performed some docking calculations as well, supposed to be shown of fig. 5B, although there was no fig 5 b in the given manuscripts (maybe fig. S10?). Moreover, no word (!) about the docking experiments could be found in the methods part.

Reply: Thanks for your comments. The method for computer modeling is added in the Materials and Methods (page 21-22).

“Molecular Modeling

Solved crystal structure of HLA-B*15:01 from the Protein Data Bank (PDB), PDB code 1XR8, were used as the basis for molecular docking HLA-B*15:02. The program SSM (Secondary-Structure Matching) in COOT was used to superimpose crystal structures and homology models⁵⁹. Initial molecular docking of the CBZ chemical to the HLA-B*15:02 structure was performed utilizing MtiOpenScreen on the Mobylye bioinformatics framework, Ressource Parisienne en BioInformatique Structurale (RPBS)⁶⁰. Then, molecular docking of the CBZ chemical with HLA-B*15:02 structure and the public $\alpha\beta$ TCR was performed using a virtual screening program AutoDock Vina⁶¹. Docking results were visualized and images generated using PyMOL (PyMOL Molecular Graphics System, Version 1.2, Schrödinger, LLC, New York, New York)…”

The results of docking experiments are shown in **Figure S12**.

Figure S12. *In silico* molecular modeling of the interaction of the public $\alpha\beta$ TCR, carbamazepine, and HLA-B*15:02 protein. The crystal structure of HLA-B*15:01 (PDB code 1XR8) was adopted for the model of HLA-B*15:02, and the molecular docking software (AutoDock Vina) was applied to predict CBZ binding to the HLA-B*15:02/peptide/TCR complex. The highest scoring interaction was identified between CBZ and a site comprised of the α 1 helix of HLA-B*15:02, TCR α CDR3 VFDNTDKLI, and TCR β CDR2 (ΔG -7.9 kcal/mol).

3-6. Then TCR transfectants were generated with a murine 5KC hybridoma background. Again the sequence and the hybridization (human-murine constant part) should be published. In figure 6, TCR transfectants were shown to be activated by carbamazepine and derivative. Although, the responses are statistically significant and the specificity of the reactivity is shown without any doubts, the reactivity seems to be quite low. According to the methods section, 105 cells were plated per well and only up to 300 cells could be activated and produced IL2, corresponding to 0.3 % of a monoclonal population. This aspect and possible explanations could be addressed in the discussion section.

Reply: The sequences of constructs for TCR transfectants are shown in **Figure S13**. Regarding the TCR transfectants experiments, we generated murine 5KC-TCR transfectants for antigen-specificity assay. As the reviewer pointed out, the frequency of reactive cells found in the IL2 ELISPOT assays was low. We think that there were different factors contributing to the observation. Our antigen-presenting cells (C1R, a human HLA class I-deficient lymphoblastoid cell line) and the effector cells (5KC, a murine T hybridoma) were xenogeneic. The murine 5KC T hybridoma had not only TCR α /TCR β deficiency, but also lacked human CD3 and CD8. The interaction between murine TCR and human MHC (i.e., HLA-B*15:02) should be much weaker than that of physiological condition. In addition, the TCR α and β chains in the 5KC-TCR hybridoma were chimeric, as we cloned the unique

sequences of human TCR α/β CDR3 regions to link to the mouse TCR constant domains by the PTV1.2A sequence (**Figure S13**). Furthermore, our public TCR was identified from human CTL, but the 5KC-TCR hybridoma lacked human CD8. These factors may lead to the low frequency of reactive cells.

3-7. Finally, the authors were able to develop a remarkable animal model for cbz induced TEN. This model functions only in presence of HLA-B*15:02 tg mice and T cells expressing the identified public TCR. These findings represent an important breakthrough in the field of DHR, which is terribly lacking useful animal models. Interestingly, the affected animal exhibited lesions not only in the skin but also in internal organs, which is more typical for DRESS.

In the submitted study, the authors focused mainly on cbz induced SJS/TEN (and oxc), for which they have the highest numbers on patients to study. For the other drugs, the identification of a public TCR shared by most patients was more complicated. For allopurinol for example, dominant clonotypes were found in 2 patients, but the clonotypes “ASSQDLTGNTI” and “ASSPRDFSYEQY” cannot be considered homologous. In the drug allergy field, people have the tendency to generalize their finding to all other hypersensitivity reactions induced by other drug. For years, haptenization has been thought to be an essential driver of DHR. When the presentation of altered peptide repertoire on HLA-B*57:01 in the presence of abacavir was discovered, every DHR was thought to be mediated by altered peptide repertoire presentation. In this study, the findings are important and represent a breakthrough for cbz induced SJS/TEN. But it is not sure yet, whether the public TCR could be generalized to other SJS/TEN caused by other drugs. This position should be discussed in the discussion.

Reply: Thanks for the comments. We added a paragraph for this issue in the **Discussion (page 16-17)**.

“To explain the interaction of HLA, drug antigen, and TCR in drug hypersensitivity, there are different hypotheses, including the “hapten” theory³⁵, the “pharmacological interaction with immune receptors (p-i)” concept², the “altered peptide repertoire” model^{36, 37}, and the “altered TCR repertoire” model³⁸. We previously applied mass-spectrometry to evaluate the peptide repertoire of HLA-B*15:02; nevertheless, no evidence suggests the presence of CBZ haptenated peptide³⁹. Our previous studies further demonstrated that the chemical antigens (e.g., CBZ or oxypurinol) could directly interact with HLA proteins without the involvement of the antigen-processing pathway, which supports the “p-i” model⁴⁰⁻⁴². In this study, we identified a predominate and public $\alpha\beta$ TCR clonotype, which can directly bind to CBZ and its structural analogs (e.g., CBZ-10,11-epoxide and OXC), and the immune response was promoted by the presence of HLA-B*15:02. The oligoclonal TCR clonotype identified in

CBZ-SJS/TEN further supports “p-i” concept, but not the “altered peptide repertoire” model, which induces polyclonal TCR^{36, 37}. These T lymphocytes with a public TCR recognizing a small chemical antigen presented by the preferred HLA molecule, may arise from the pre-existing memory T cells by heterologous immune response⁴. Whether the public TCR could be generalized to other SJS/TEN caused by other drugs needs further studies.”

3-6. Altogether, this study represents a real breakthrough because of two important aspects. First a public TCR could be identified in patients of different ethnicities (HLA) with cbz induced SJS/TEN. It was thought for very long time, that particular TCR sequences in the TCR repertoire might represent a risk for the development of severe DHR. Here the authors could prove this assumption for the first time. Secondly, an animal model for SJS/TEN could be generated that is at last not based on the hapten theory.

Reply: Thanks for the comments.

Other points:

3-7. - Suppl table 1

Cbz induced TEN in Caucasian have been associated with the allele HLA-A*31:01. If the alleles of the HLA-A and C are also known, it would be important to communicate them.

Reply: Thanks. We’ve the information of HLA-A data in the **Table S1**.

The HLA-A, HLA-B, and HLA-C genotypes of the 13 cases of the European patients with CBZ-SJS/TEN are shown below.

HLA-A, HLA-B, and HLA-C genotypes of the 13 cases of the European patients with CBZ-SJS/TEN

Patient ID	Culprit drug	Clinical diagnosis	HLA-A genotype	HLA-B genotype	HLA-C genotype	Populations
Case25	CBZ	SJS	A*01:01 / A*66:01	B*07:02 / B*41:02	C*07:01 / C*17:01	Europeans
Case26	CBZ	SJS	A*02:01 / A*11:01	B*15:02 / B*44:02	C*05:01 / C*08:01	Asian descent
Case27	CBZ	SJS	A*02:01 / A*23:01	B*41:01 / B*49:01	C*07:01 / C*07:01	Europeans
Case28	CBZ	SJS	A*02:01 / A*11:01	B*15:02 / B*44:02	C*05:01 / C*08:01	Asian descent
Case29	CBZ	SJS	A*02:07 / A*34:01	B*15:21 / B*46:01	n.a.	Asian descent
Case30	CBZ	SJS	A*01:01 / A*03:01	B*08:01 / B*35:03	C*04:01 / C*07:01	Europeans
Case31	CBZ	SJS	A*01:01 / A*01:01	B*08:01 / B*57:01	C*06:02 / C*07:01	Europeans
Case32	CBZ	SJS	A*24:02 / A*26:01	B*35:01 / B*38:01	n.a.	Europeans
Case33	CBZ	SJS	A*02:01 / A*03:01	B*07:02 / B*57:01	C*06:02 / C*07:02	Europeans
Case34	CBZ	SJS	A*01:01 / A*31:01	B*57:01 / B*57:01	C*06:02 / C*07:01	Europeans
Case35	CBZ	SJS	A*01:01 / A*30:02	B*08:01 / B*55:01	C*04:01 / C*07:01	Europeans
Case36	CBZ	SJS	A*01:01 / A*31:01	B*13:01 / B*57:01	C*05:01 / C*06:02	Europeans
Case37	CBZ	SJS	A*01:01 / A*66:01	B*08:01 / B*41:02	n.a.	Europeans

3-8. -Figure 2

The figure is not clear enough. The letters are barely readable.

Reply: Thank for the comments. We've revised the Figure 2.

3-9. - Figure S7

Please notify the color of the α and β chains. Is the pink segment coming from the α or from the β chain?

Reply: Thank for the comments. We've revised the figure legends.

Figure S6. The TCR α and TCR β repertoire of the blister cells of a representative patient with CBZ-SJS/TEN. The TCR α and TCR β repertoire of blister cells of a representative patient with CBZ-SJS/TEN (case 4) were analyzed by high-throughput next-generation sequencing. **(a)** The five most common TCR β CDR3 clonotypes (shown in orange) and TCR α CDR3 clonotypes (shown in pink), and their corresponding variable/joining types and respective frequencies are listed. **(b)** The paired TCR α and TCR β clonotypes were suggested by the order of frequency ranking, and the most common TCR α CDR3 clonotype “VFDNTDKLI” and TCR β CDR3 clonotype “ASSLAGELF” accounted for 44.81% and 42.85% of total reads of TCR α and TCR β CDR3 clonotypes, respectively, were paired.

3-10. - Figure 5

Section B is missing, as well as the docking part in the method section

Reply: Thanks for your comments. The method for computer modeling is added in the Materials and Methods (page 21-22).

The results of docking experiments are shown in **Figure S12**.

3-11. - Soluble TCR and TCR transfectants

Please provide the complete sequence of all the constructs.

Reply: As suggested, the complete sequence of soluble single-chain TCR is shown in **Figure S8**. The complete sequence of TCR transfectants was shown in **Figure S13**.

3-12. - Lane 67-68. The sentence is not grammatically correct

Reply: Thank you. We've revised the sentences.

REVIEWERS' COMMENTS:

Reviewer #1 (Remarks to the Author):

The authors have replied to my concerns and added relevant material to the supplementary section, I recommend the manuscript to be published in Nature Communications.

Reviewer #2 has not left comments to the authors but recommends publication in the notes to editors, and emphasizes the importance to deposit TCR sequence data in a public database such as SRA, unless there are legal or ethical restrictions.

Reviewer #3 (Remarks to the Author):

The manuscript has been revised and it gained in quality. Most of my major and minor concerns have been addressed. Besides many data have been provided in the form of supplementary figures and tables, like complete HLA genotypes, TCR sequences, docking values... In the revised manuscript, I noticed that the figure 1a has been mistakenly modified. Indeed the table of the right hand side does not correspond any more to the previous version and most importantly, the data do not reflect an overexpression of TRBV 12-4 in the CBZ or Ox-CBZ patients. In the row data excel file, values seem to correspond to the original version. Please correct this, since this figure represents the basement of the entire study. Otherwise, I don't have other objection any more.

REVIEWERS' COMMENTS:

Reviewer #1 (Remarks to the Author): The authors have replied to my concerns and added relevant material to the supplementary section, I recommend the manuscript to be published in Nature Communications.

Reply: Thanks for reviewer's comment.

Reviewer #2 has not left comments to the authors but recommends publication in the notes to editors, and emphasizes the importance to deposit TCR sequence data in a public database such as SRA, unless there are legal or ethical restrictions.

Reply: Thanks for reviewer's comment. As described in Data availability section, the sequence data that support the findings of this study have been deposited in the NCBI sequence read archive (SRA) database with links to BioProject accession number "PRJNA550004".

Reviewer #3 (Remarks to the Author): 13 The manuscript has been revised and it gained in quality. Most of my major and minor concerns have been addressed. Besides many data have been provided in the form of supplementary figures and tables, like complete HLA genotypes, TCR sequences, docking values... In the revised manuscript, I noticed that the figure 1a has been mistakenly modified. Indeed the table of the right hand side does not correspond any more to the previous version and most importantly, the data do not reflect an overexpression of TRBV 12-4 in the CBZ or Ox-CBZ patients. In the row data excel file, values seem to correspond to the original version. Please correct this, since this figure represents the basement of the entire study. Otherwise, I don't have other objection any more.

Reply: Thanks for reviewer's comment. We have corrected the figure 1a in the final version of the manuscript.